# Continuous signaling of CD79b and CD19 is required for the fitness of Burkitt lymphoma B cells

Xiaocui He[1,2] (iD), Kathrin Kläsener[1,2] (iD), Joseena M Iype[1,†], Martin Becker[1,‡] (iD), Palash C Maity[1,†], Marco Cavallari[1], Peter J Nielsen[2], Jianying Yang[1,2] (iD) & Michael Reth[1,2,*] (iD)

## Abstract

**Expression of the B-cell antigen receptor (BCR) is essential not only for the development but also for the maintenance of mature B cells. Similarly, many B-cell lymphomas, including Burkitt lymphoma (BL), require continuous BCR signaling for their tumor growth. This growth is driven by immunoreceptor tyrosine-based activation motif (ITAM) and PI3 kinase (PI3K) signaling. Here, we employ CRISPR/Cas9 to delete BCR and B-cell co-receptor genes in the human BL cell line Ramos. We find that Ramos B cells require the expression of the BCR signaling component Igβ (CD79b), and the co-receptor CD19, for their fitness and competitive growth in culture. Furthermore, we show that in the absence of any other BCR component, Igβ can be expressed on the B-cell surface, where it is found in close proximity to CD19 and signals in an ITAM-dependent manner. These data suggest that Igβ and CD19 are part of an alternative B-cell signaling module that use continuous ITAM/PI3K signaling to promote the survival of B lymphoma and normal B cells.**

**Keywords** B-cell antigen receptor; Burkitt lymphoma; Cas9; CRISPR; survival signal; tumor fitness
**Subject Categories** Immunology; Signal Transduction
**The EMBO Journal (2018) 37: e97980**

## Introduction

The B-cell antigen receptor (BCR) consists of the membrane-bound immunoglobulin (mIg) comprising two heavy (H) and two light (L) chains and the signal-transducing Igα/Igβ (CD79a/CD79b) heterodimer (Reth & Wienands, 1997). The cytoplasmic tails of Igα and Igβ each contain an immunoreceptor tyrosine-based activation motif (ITAM) with two conserved tyrosines that are crucial for the development and maintenance of mature B cells (Reth, 1989; Torres *et al*, 1996; Kraus *et al*, 2001, 2004; Reichlin *et al*, 2001). Upon antigen ligation and opening of the BCR, the spleen tyrosine kinase (Syk) phosphorylates and interacts with the ITAM tyrosines of Igα and Igβ (Rolli *et al*, 2002). The resulting ITAM/Syk complex amplifies the BCR signal and connects the BCR to several downstream signaling pathways, leading to the activation, proliferation, and differentiation of B cells (Johnson *et al*, 1995; Kurosaki, 1999; Deane & Fruman, 2004; Werner *et al*, 2010). A prominent substrate of Syk is the SLP65/BLNK adaptor protein which, upon phosphorylation, organizes the B-cell calcium signalosome and thus is required for calcium mobilization in activated B cells (Jumaa *et al*, 1999; Kulathu *et al*, 2008). Another important signaling hub in B cells is the CD19 co-receptor molecule, which forms a complex on the plasma membrane together with the tetraspanners, CD81 (TAPA-1) and leu-13, and the complement receptor CD21 (CR2; Fearon & Carroll, 2000). The long cytosolic tail of CD19 contains nine tyrosines most of which are phosphorylated by the src-family kinase Lyn upon B-cell activation. Once phosphorylated, these tyrosines serve as binding partners for the adaptor proteins Grb2 and Vav, the phosphoinositide-3 kinase (PI3K), and phospholipase C-γ (Fearon & Carroll, 2000). Signal transduction from CD19 thus involves PI3K signaling and cytoskeleton rearrangements.

Mature B cells co-express two different classes of BCR, namely the IgM-BCR and the IgD-BCR, which reside in different protein islands on the plasma membrane (Yang & Reth, 2010; Maity *et al*, 2015). On resting B cells, CD19 complex and the CD81-tetraspanin complex are localized inside IgD-class-specific protein islands (Mattila *et al*, 2013; Maity *et al*, 2015). On activated B cells, however, the CD19 complex is found in close proximity to the open IgM-BCR and thus gains access to ITAM signaling (Klasener *et al*, 2014). The IgM/CD19/CD81/CD21 complex is also stabilized by its co-ligation with complement-bound antigens that can reduce the threshold for BCR signaling 10- to 100-folds (Carter & Fearon, 1992). The mouse CD19-deficient B cells have defects in proliferation and maturation in peripheral lymph tissues and spleen, and have a defective T-cell dependent antigen response (Rickert *et al*, 1995).

The expression of a pre-BCR and a BCR is required for B-cell development and the maintenance of mature B cells. Mice with a deletion of any one of the H chain, Igα, or Igβ genes display a

1 BIOSS Centre For Biological Signaling Studies, Department of Molecular Immunology, Biology III, Faculty of Biology, University of Freiburg, Freiburg, Germany
2 Max Planck Institute of Immunobiology and Epigenetics, Freiburg, Germany
*Corresponding author. Tel: +49 761 203 97663; E-mail: michael.reth@bioss.uni-freiburg.de
†Present address: Institute for Immunology, Uni Hospital Ulm, Ulm, Germany
‡Present address: Helmholtz Zentrum München, München, Germany

developmental block at the pro-B-cell stage (Kitamura *et al*, 1991; Gong & Nussenzweig, 1996; Torres *et al*, 1996; Minegishi *et al*, 1999; Reichlin *et al*, 2001; Meffre & Nussenzweig, 2002; Pelanda *et al*, 2002). The deletion of the H chain gene or the mutation of the ITAM tyrosines of Igα in mature B-cell results in apoptosis and disappearance of B cells from the periphery in a few days (Lam *et al*, 1997; Kraus *et al*, 2004). These data show that the proper expression of the BCR is required for the fitness and survival of mature B cells and it was therefore suggested that the unengaged BCR emits a maintenance or tonic survival signal. However, some mature B cells which loose BCR expression after an inducible deletion of the Igα gene survive in mice for more than 20 days, suggesting alternative signals for the survival of these B cells (Levit-Zerdoun *et al*, 2016).

Expression of, and signaling from, the pre-BCR or BCR is also required for the continuous growth of several B-cell lymphomas (Gauld *et al*, 2002; Kuppers, 2005; Lenz & Staudt, 2010; Stevenson *et al*, 2011). For instance, most B cells of chronic lymphocytic leukemia (B-CLL) carry an auto-aggregated BCR, which promotes continuous ITAM and PI3K signaling and is required for the expansion and survival of these tumor cells (Duhren-von Minden *et al*, 2012). Activated B-cell-like diffuse large B-cell lymphomas (ABC-DLBCL) carry mutations of the BCR and its signal components, resulting in chronic active NFκB signaling and increased tumor survival (Davis *et al*, 2010). Burkitt lymphoma (BL) is an aggressive B-cell tumor that is derived from germinal center B cells and is transformed by a translocation of the c-Myc oncogene into the H chain locus (Dalla-Favera *et al*, 1982). Knock-down experiments and the application of specific inhibitors suggest that the survival and expansion of BL cells require ITAM and PI3K signaling (Schmitz *et al*, 2012, 2014). Furthermore, a recent study on a BL mouse model suggests that the expression of the BCR is required for the fitness of these tumor cells (Varano *et al*, 2017). Here, we have used the CRISPR/Cas9 method to delete genes that code for the receptor and signaling components of the BCR in the human BL cell line Ramos. A competitive growth assay showed that the fitness of the Ramos B cells requires the expression of the BCR co-receptor CD19 and of the BCR subunit Igβ. We found that Igβ is expressed in a mIg-independent fashion on the B-cell plasma membrane where it is found in close proximity to CD19. We suggest that Igβ and CD19 are part of an alternative receptor module for tonic ITAM and PI3K signaling that promotes the survival of normal and tumor B cells.

## Results

### The fitness of Ramos cells depends on the mIg-independent expression of Igβ

To study the role of the BCR complex in supporting the growth of B-cell lymphomas, we used the CRISPR/Cas9 gene-editing tool to generate Ramos B-cell mutants lacking all (BCR-null) or some of the four BCR components, H and L chains, Igα and Igβ, (Fig 1A). We then compared the proliferation of the BCR-null line with that of wild-type (WT) Ramos cells (Fig 1B). Surprisingly, the BCR-null cells expanded in culture with similar kinetics to that of WT Ramos B cells, indicating that they were not compromised in their *in vitro* growth (Fig 1B). We next compared the two Ramos cell lines in a

growth competition assay (Fig 1C) in which we labeled WT Ramos B cells with GFP and mixed them in a 1:1 ratio with either unstained (GFP⁻) WT or BCR mutant Ramos cells. A FACScan analysis verified that BCR-null Ramos cells did not carry any BCR (neither the IgM-BCR nor the IgD-BCR) on their cell surface (Fig 1D). Interestingly, in the growth competition assay, the BCR-null Ramos cells gradually disappear from the culture within 8 days, thereby suggesting that they are less fit than WT Ramos B cells (Fig 1E).

We next asked whether the loss of one or both of the signaling components of the BCR caused the reduced fitness of the BCR-null B cells. For this, we generated Igα and Igβ single KO as well as Igα/Igβ double KO Ramos B cells and verified in a FACScan analysis that none of these cells carry an IgM-BCR or IgD-BCR on their cell surface (Fig 1F). The loss of the BCR signaling component was also verified by an intra- and extracellular FACScan as well as by Western blot analysis (Appendix Fig S1). When the different single or double KO cells were cultured separately, they expanded as well as the WT Ramos B cells (Appendix Fig S2). However, in the competition with WT Ramos B cells, only the Igβ single or Igα/Igβ double KO cells were lost within 8 days of culture whereas the Igα single KO cells competed well with the WT Ramos B cells (Fig 1G). To exclude that these results are influenced by the generation of variants during the selection and prolonged culture of Ramos KO clones, we repeated the competition experiments with a new set of Ramos KO clones that were culture for only 14 days and obtained the same results (Appendix Fig S3). Together, this analysis suggests that the fitness of the Ramos B cells is not associated with total BCR but rather with Igβ expression.

We next analyzed whether the apparent signaling function of Igβ on Ramos B cell requires the expression of the mIg part of the BCR. We generated HL double as well as HLα and HLβ triple KO Ramos BCR mutant cell lines (Fig 2A) and verified the loss of the corresponding BCR components by a FACScan and a Western blot analysis (Appendix Fig S1D and E). Interestingly, the FACScan analysis using anti-Igβ antibodies showed that the HLα triple KO Ramos B cells still expressed some Igβ protein on their cell surface albeit in reduced amounts in comparison with WT Ramos B cells (Fig 2B). In the competition assay with WT Ramos cells, only the HLβ KO cells lacking Igβ expression gradually disappear from the culture (Fig 2C). The finding that the Igβ protein is expressed in the absence of mIg on the surface of HLα KO Ramos B cell prompted us to test whether anti-Igβ antibodies could stimulate these cells. We thus exposed WT, HL-KO, HLα-KO, and HLβ-KO Ramos cells to the monoclonal anti-Igβ antibody and measured the calcium mobilization in these cells by FACScan (Fig 2D–G). This analysis shows that Ramos cells expressing Igβ in the absence of mIg can be stimulated by anti-Igβ antibody, albeit in a delayed and reduced manner compared with the WT Ramos B cells. The HLβ-KO triple KO Ramos cells did not respond to the anti-Igβ antibody treatment and served as negative control in this experiment (Fig 2G). As a control, we exposed the four different Ramos clones to anti-Igα antibodies and found that only the WT Ramos B cells responded (Appendix Fig S4).

### A functional ITAM is required for Igβ-dependent B-cell fitness

The cytoplasmic tail of human Igβ carries two conserved ITAM tyrosines (Y196 and Y207) that are important for the interaction of the BCR signaling components with Syk. To test whether the fitness

    

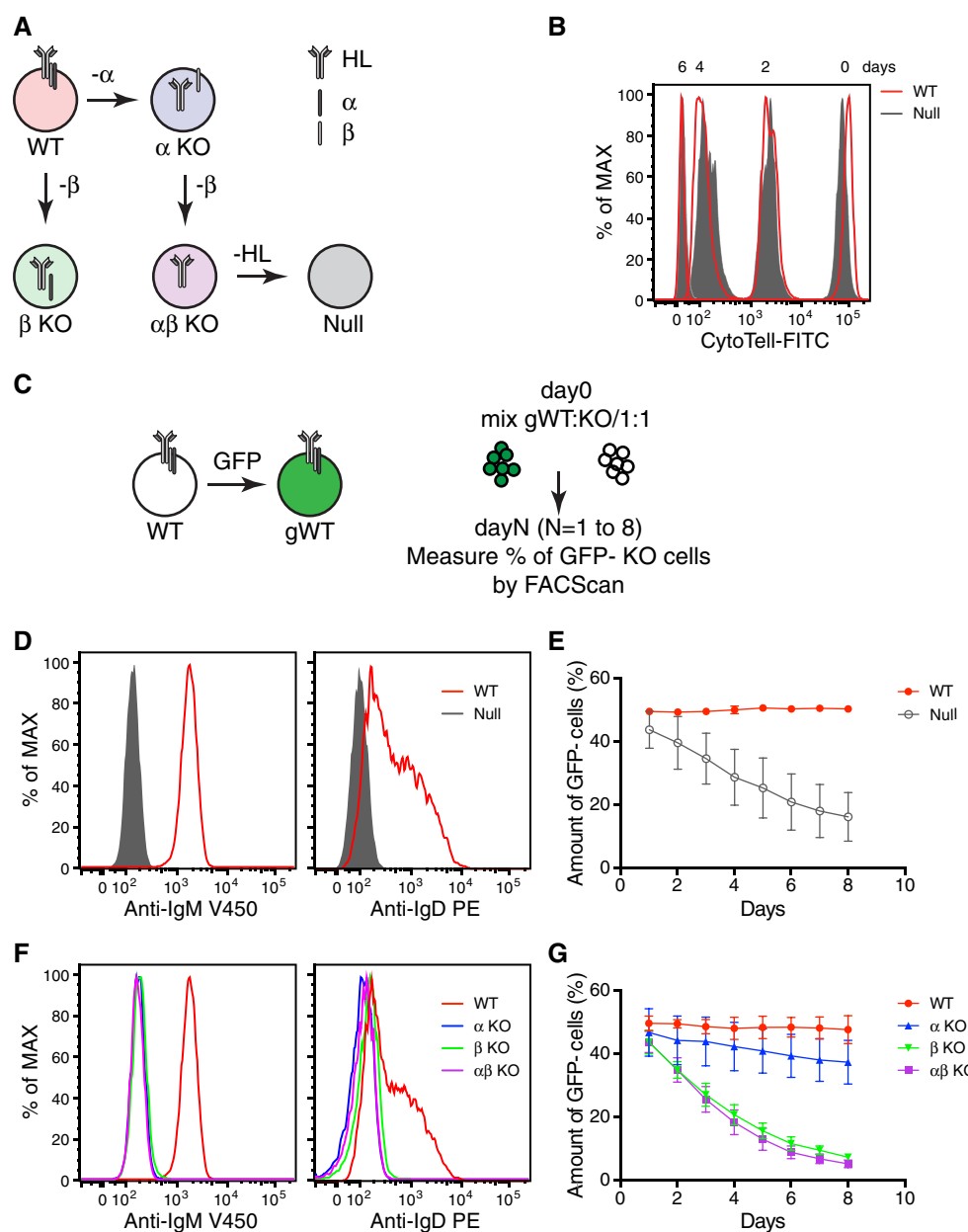

**Figure 1.  Fitness of Ramos cells depends on the Igβ subunit of the BCR.**

A   A schematic diagram showing the route for the generation of single- and multi-BCR components KO from wild-type (WT) Ramos B cells by the CRISPR/Cas9 method.

B   Cell proliferation assay of WT and Ramos-null cells using CytoTell™ UltraGreen. Null = HC/LC/Igα/Igβ tetra KO.

C   A diagram depicting the growth competition assay. WT Ramos cells were retrovirally transduced with pMIG empty vector (EV) to create GFP-labeled WT cells (gWT). BCR component KO cells were mixed together with gWT Ramos cells at about 1:1 ratio at day 0, and the relative amount of GFP⁻ BCR components KO cells was then measured by flow cytometry at different time points.

D   The expression levels of IgM-BCR and IgD-BCR on the surface of WT and Ramos-null cells were determined by flow cytometry.

E   Growth competition between Ramos-null and gWT cells. The competition between GFP⁻ WT and gWT cells serves as a control. The data represent the mean and standard error of a minimum of three independent experiments.

F   The expression levels of IgM-BCR and IgD-BCR on the surface of WT and BCR component KO Ramos cells were determined by flow cytometry.

G   Growth competition of the BCR components KO cells against the gWT cells.

Data represent the mean and standard error of a minimum of three independent experiments. One clone is used for each genotype.

of Ramos B cells depends on ITAM signaling, we transduced HLβ KO Ramos B cells with either empty vector (EV) or with vectors encoding WT or ITAM-mutated Igβ proteins and verified the expression of the Igβ proteins on the surface of the transduced HLβ KO cells by FACScan (Fig 3A). In co-cultures containing transduced and non-transduced HLβ KO cells, the Igβ-producing Ramos cells

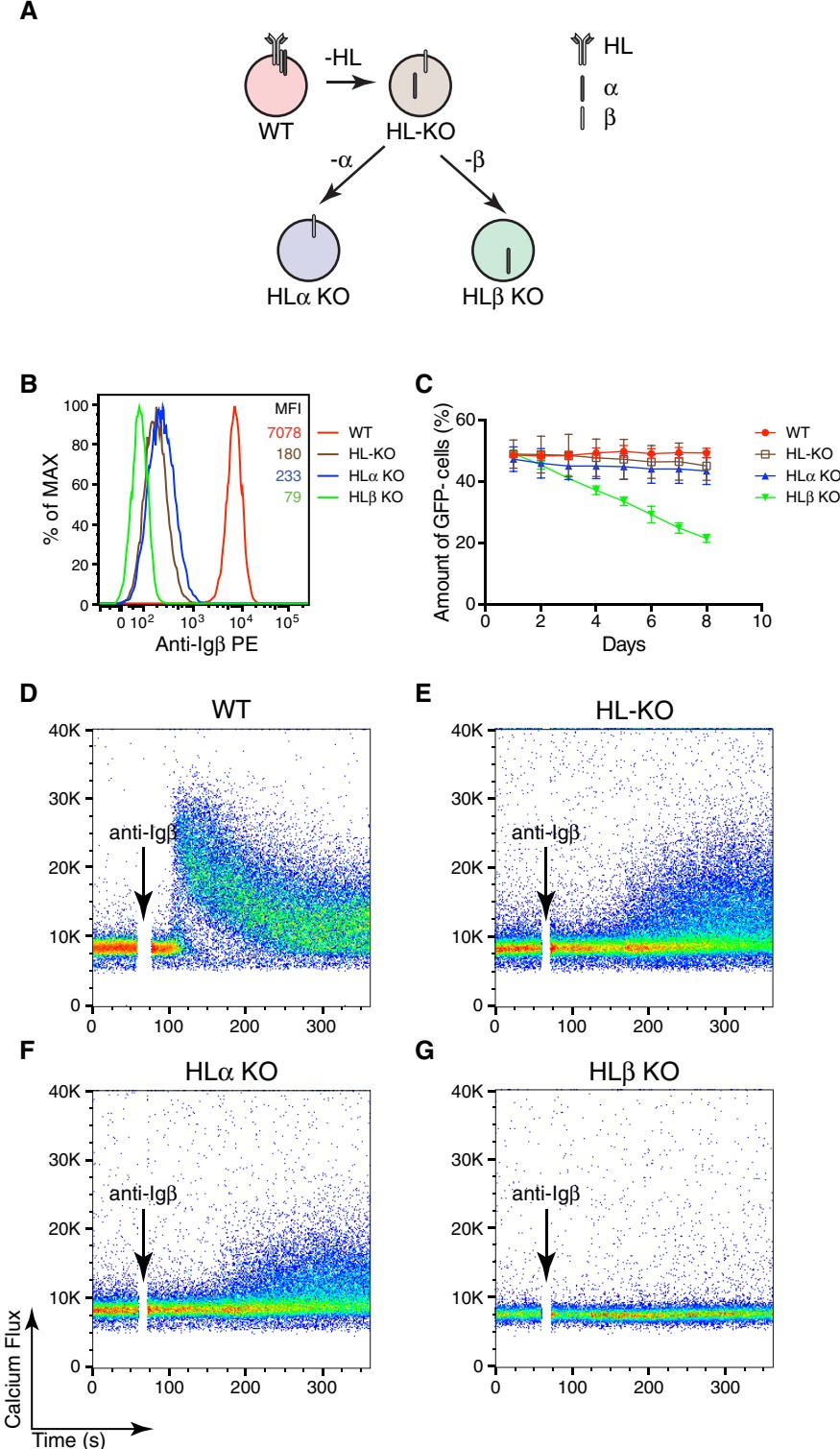

**Figure 2. BCR-independent Igβ expression determines the fitness of Ramos cells.**

A A schematic diagram showing the route map for generating single- and multi- BCR component KOs from WT Ramos B cells by the CRISPR/Cas9 method.

B The expression of Igβ on the surface of different BCR component KO Ramos cells was determined by flow cytometry.

C Growth competition of the BCR component KO cells against gWT cells. The data represent the mean and standard error of a minimum of three independent experiments.

D–G Calcium responses of WT and BCR component KO Ramos cells upon the stimulation with anti-Igβ antibodies. HL-KO = HC/LC double KO; HLα KO = HC/LC, Igα triple KO; HLβ KO = HC/LC, Igβ triple KO. One clone is used for each genotype.

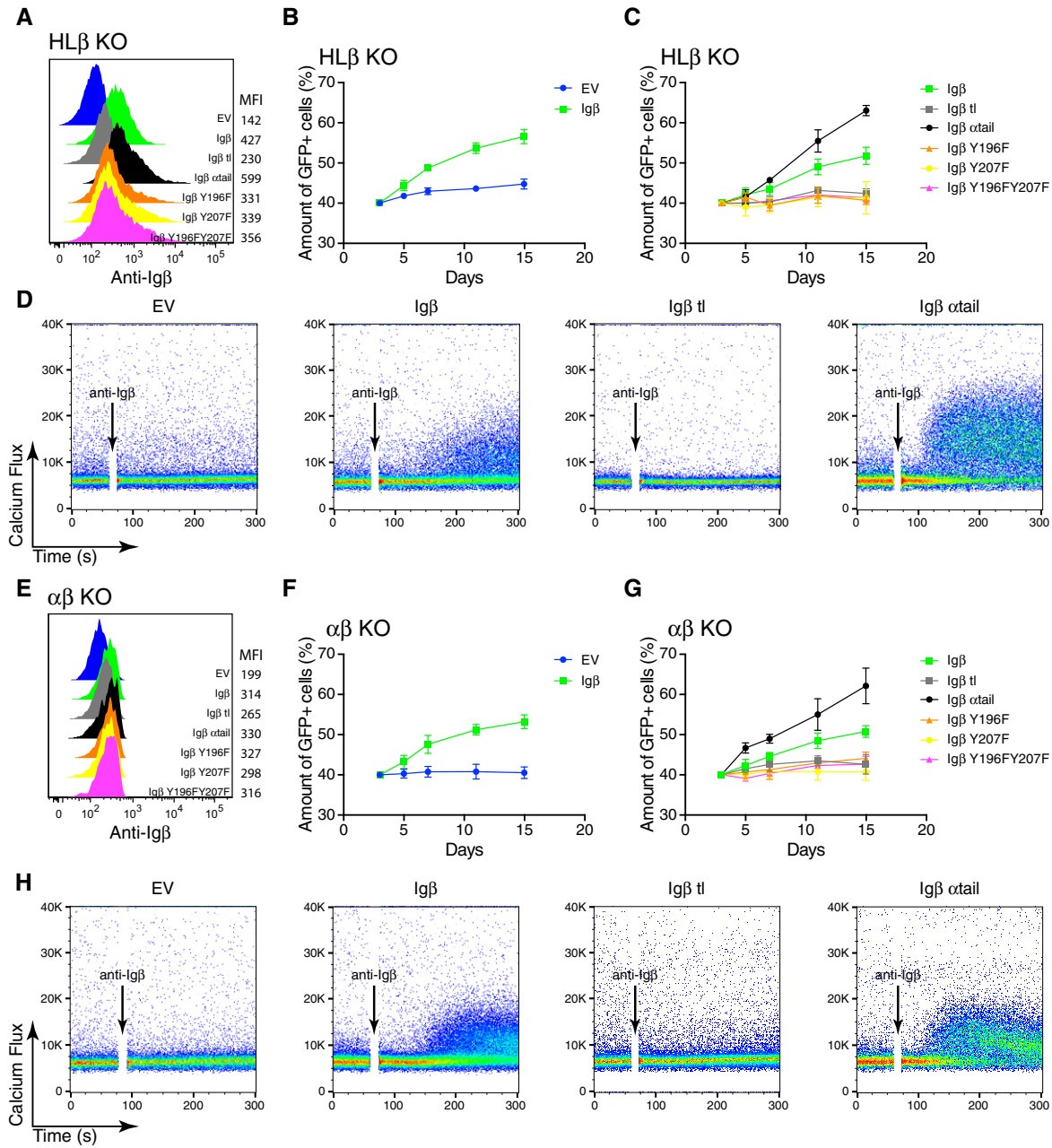

**Figure 3. Signaling through the ITAM of BCR-independent Igβ contributes to the fitness of Ramos cells.**

A    The expression of Igβ on the surface of HLβ KO cells reconstituted with WT or different Igβ mutants was determined by flow cytometry.

B, C    The percentage of GFP-positive transduced cells at different time points after the transduction of HLβ KO cells reconstituted with WT or different mutated form of Igβ is shown. The vectors express GFP as a transduction marker. The data represent the mean and standard error of a minimum of three independent experiments.

D    Calcium responses of HLβ KO cells reconstituted with WT or different Igβ mutant constructs upon the stimulation of anti-Igβ antibodies. The data are representative of three independent experiments.

E    The expression of Igβ on the surface of αβ KO cells reconstituted with WT and different Igβ mutant constructs was determined by flow cytometry.

F, G    The proportion of GFP-positive αβ KO Ramos cells at different time points after their transduction with vectors encoding either WT or different mutated forms of Igβ. The vectors express GFP as a transduction marker. The data represent the mean and standard error of a minimum of three independent experiments.

H    The calcium responses of αβ KO cells reconstituted with WT or different Igβ mutant constructs stimulated by anti-Igβ antibodies. The data are representative of three independent experiments. A minimum of two clones were used for both the HLβ KO and the αβ KO.

were enriched over time whereas the EV transfected control cells did not show this effect (Fig 3B). The growth-promoting effect of Igβ was abolished, however, when one or both ITAM tyrosines were mutated to phenylalanine (Fig 3C). Thus, the fitness of Ramos B cells depends on the ITAM signaling mechanism. This conclusion was also confirmed by the analysis of HLβ KO cells expressing an

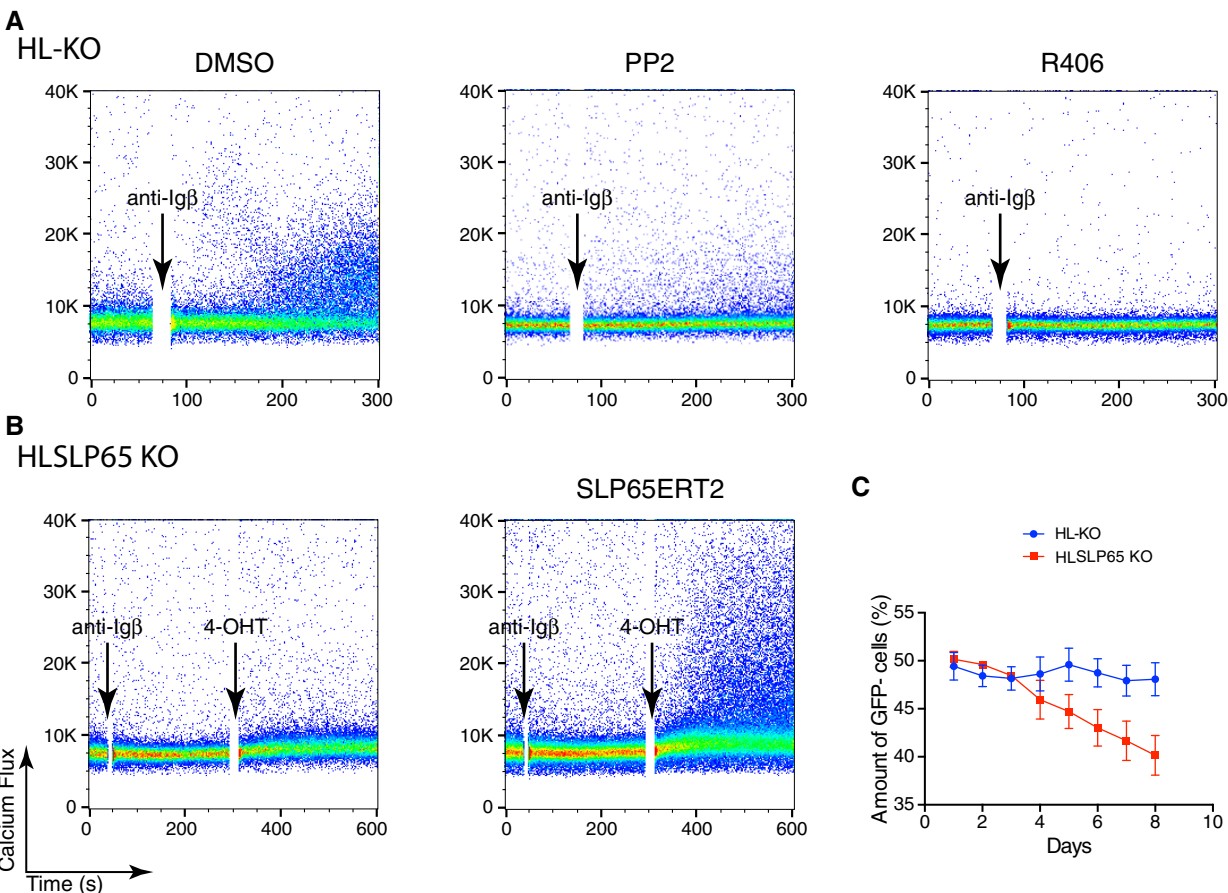

**Figure 4. The BCR-independent Igβ signals though the canonical BCR signaling pathway.**

A   Calcium responses of HL double KO Ramos cells stimulated by anti-Igβ after treatment of the indicated kinase inhibitors. The data are representative of three independent experiments.

B   Calcium responses of HLSLP65 triple KO Ramos cells reconstituted with a tamoxifen-inducible SLP65 after the stimulation with anti-Igβ and treatment with 4-OHT. Cells without SLP65 production were used as controls. The data are representative of three independent experiments.

C   Growth competition assay of HLSLP65 KO cells with HL-KO cells. HLSLP65 KO cells were mixed together with GFP-expressing HL-KO cells. The relative amount of HLSLP65 KO cells was then measured at the indicated time points by flow cytometry. The data represent the mean and standard error of a minimum of three independent experiments. One clone is used for each genotype.

Igβ-α tail chimera, which expressed the extracellular portion and transmembrane region of Igβ and the cytosolic tail of Igα. ITAM signaling from this chimeric receptor provided an even higher growth advantage, whereas an Igβ tail truncation (Igβtl) could not increase the fitness of HLβ KO Ramos B cells (Fig 3C).

We next exposed some of the transduced HLβ KO Ramos B cells to the monoclonal anti-Igβ antibody and measured the calcium mobilization in these cells by FACScan (Fig 3D). The HLβ KO Ramos B cells expressing WT Igβ or the Igβ-α tail chimera displayed an increased calcium mobilization whereas EV control or Igβ tail-less expressing HLβ KO Ramos B cells did not respond to this stimulus. This analysis shows that the increased fitness of the Ramos B cells is correlated with increased responsiveness toward anti-Igβ antibody stimulation. The above analysis was repeated with αβ KO Ramos B cells with the same results (Fig 3E–H). Thus, it is the expression of Igβ and not the formation of an Igα/Igβ heterodimer that is required for the increased fitness, and the anti-Igβ induced calcium release of Ramos B cells. Taken together, these results

suggest that Igβ promotes cellular fitness by a constant signaling through its ITAM tyrosines.

**Igβ employs the same signaling pathway as the activated BCR**

Upon exposure to its cognate antigen, the BCR interacts with the kinases Lyn and Syk, both of which support the dissociation and activation of the BCR and mediate downstream signaling (Klasener *et al*, 2014). To test whether Igβ also uses these classical BCR signaling components, we pre-treated HL-KO Ramos B cells with either DMSO, the Lyn inhibitor PP2, or the Syk inhibitor R406 and then measured their calcium response after anti-Igβ stimulation. We found that the calcium response was most strongly reduced by the Syk inhibitor and to a lesser extent by the Lyn inhibitor (Fig 4A). This result demonstrates that Syk and Lyn are involved in the Igβ-mediated signaling. Indeed when we exposed HL or HLβ KO Ramos B to the oxidant pervanadate, we found that the latter cells lacking Igβ contain reduced pSyk levels (Fig EV1).

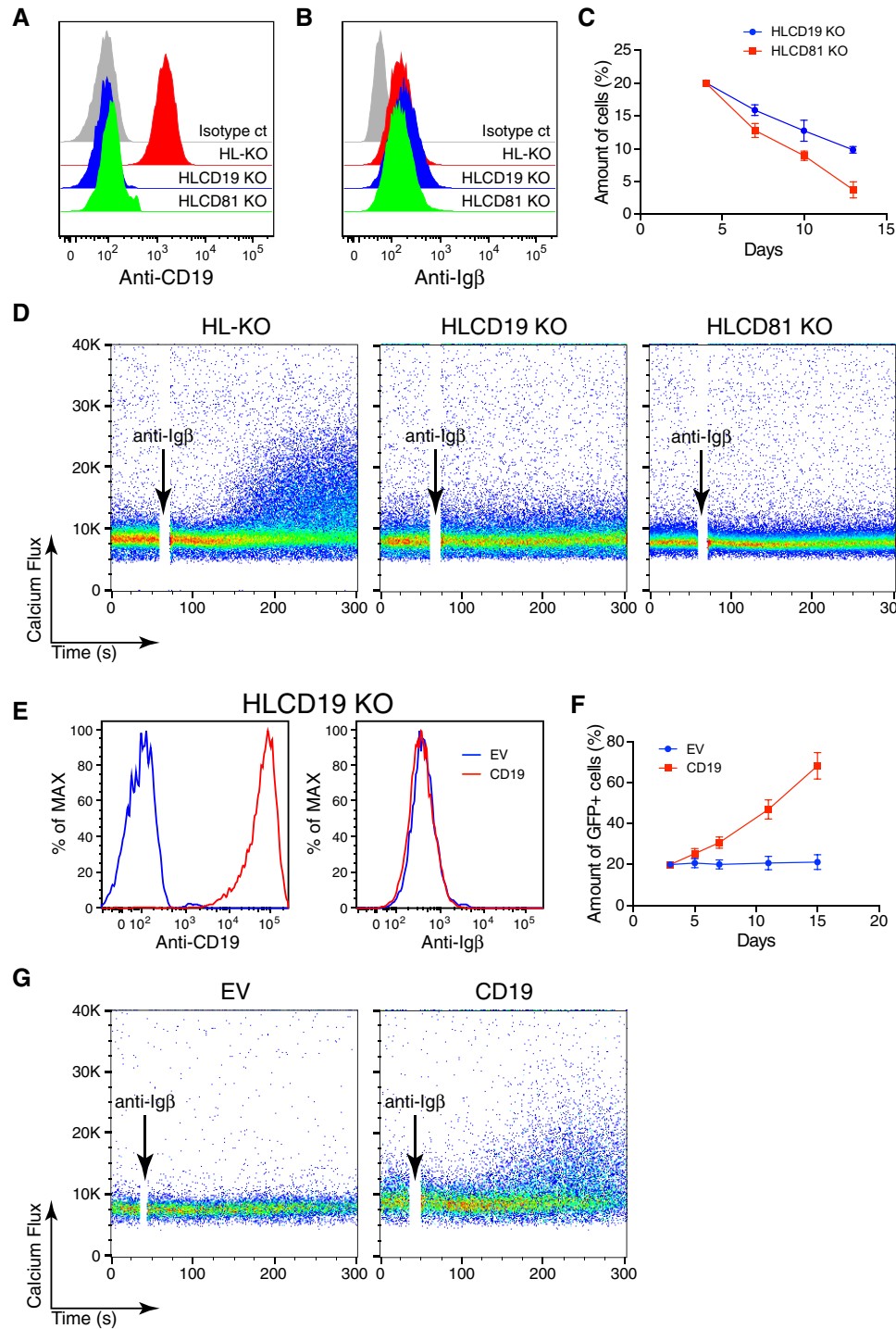

**Figure 5.  The Igβ-dependent Ramos cell fitness requires CD19.**

A, B  The expression of CD19 and Igβ on the surface of different Ramos KO cells was determined by flow cytometry.

C  The CD19 and CD81 genes in HL-KO Ramos cells were rendered defective by the CRISPR/Cas9 method, and the percentage of the CD19- or CD81-negative cells in the triple KO population was measured by flow cytometry at the indicated time points. The data represent the mean and standard error of three independent experiments.

D  Calcium responses of HL-KO, HLCD19 KO, and HLCD81 KO cells stimulated by anti-Igβ antibodies. The data are representative of three independent experiments.

E  Expression of CD19 or Igβ on the surface of HLCD19 KO cells transduced with empty vector (EV) or the CD19 vector measured by flow cytometry.

F  Percentage of GFP-positive HLCD19KO cells at different time points after their transduction with EV or the CD19 expression vector. The data represent the mean and standard error of three independent experiments.

G  Calcium responses of HLCD19 KO cells transduced with EV or CD19 expression vector after their stimulation with anti-Igβ antibodies. The data are representative of three independent experiments. One clone is used for HL-KO. HLCD19KO and HLCD81 KO are batch sorted.

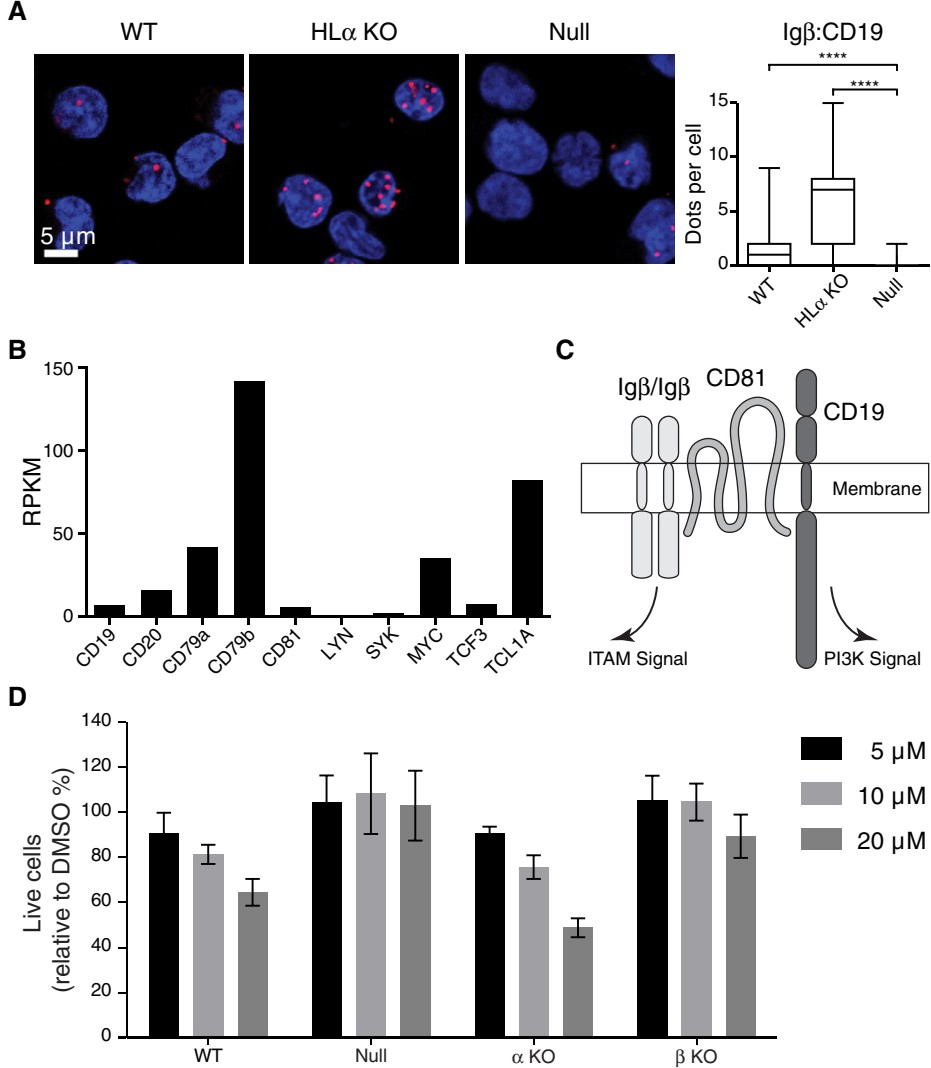

**Figure 6. Increased CD19/Igβ proximity on BCR-negative Ramos cells.**

A   Fab-PLA of Igβ:CD19 proximity in WT, HLα KO, and null Ramos cells. Data are representative of three independent experiments. *P*-values were calculated by non-parametric Mann–Whitney test. **** indicates *P* < 0.0001. In the box plot, horizontal lines in the middle represents median, box limits represent 25% to 75%, and lower and upper whisker represent min. and max. data range respectively.

B   mRNA expression levels of representative genes calculated from Ramos RNA-seq data (unpublished).

C   Model of the Igβ/CD19 module promoting ITAM and PI3K signaling.

D   The proliferation of WT, null, α KO, and β KO Ramos cells treated with different doses of the PI3K inhibitor CAL-101. DMSO treatment was used as a control. Viable cells were measured 4 days post-treatment by flow cytometry. The data represent the mean and standard error of a minimum of three independent experiments. One clone is used for each genotype.

One prominent substrate of Syk is the adaptor protein SLP65/BLNK, a component of the calcium signalosome of B cells (Fu *et al*, 1998; Wienands *et al*, 1998). To test whether SLP65 is also required for Igβ signaling, we generated HLSLP65 triple KO Ramos B cells and verified the loss of SLP65 expression by Western blot (Appendix Fig S5). The cells were then transduced with plasmids encoding SLP65ERT2, a tamoxifen-inducible form of SLP65 (Trageser *et al*, 2009). Upon exposure to anti-Igβ and tamoxifen, the SLP65ERT2-transduced HLSLP65 triple KO Ramos cells showed a calcium flux, whereas cells lacking SLP65ERT2 display no change in the calcium level (Fig 4B). This indicates that SLP65 expression is required for

proper Igβ signaling. In agreement with this conclusion, we found that the HLSLP65 triple KO cells show a growth disadvantage when co-cultured with HL-KO B cells (Fig 4C). Together, these results show that Igβ improves the fitness of Ramos B cells by employing the same Lyn, Syk, and SLP65 signaling route as activated B cells.

**The co-receptor CD19 is required for the Igβ-induced fitness signal**

All mature B cells express the co-receptor CD19, which requires association with the tetraspanner CD81 for its stable expression

on the B-cell surface (Maecker & Levy, 1997). Upon B-cell activation, CD19 and the IgM-BCR move closer together and both receptors cooperate in the amplification of the BCR signal (Fujimoto *et al*, 2000; Klasener *et al*, 2014). To test for the role of CD19 in Igβ signaling, we generated by bulk transfection a HLCD19 or HLCD81 triple KO Ramos population (Appendix Fig S6) and verified by a FACScan analysis that these cells had lost CD19 but maintained Igβ expression on their cell surface (Fig 5A and B). After transfection of the HL double KO cells with the CRISPR/Cas9 KO vector for either CD19 or CD81, the surface CD19-negative Ramos B cells gradually disappeared from the culture within 12 days (Fig 5C). Furthermore, the HLCD19 and HLCD81 mutant Ramos cells were also unable to mount a calcium response upon exposure to anti-Igβ antibodies (Fig 5D). This indicates that the expression of CD19 is required for Igβ signaling and the fitness of Ramos B cells.

We next verified the fitness-promoting role of CD19 by a gain-of-function assay. We retrovirally transduced the HLCD19 triple KO cells with either the EV control or the CD19 expression vector. The latter cells show large amounts of CD19 on their surface while the amount of surface Igβ is the same in EV and CD19-transduced HLCD19 KO cells (Fig 5E). Compared to EV, the CD19 (GFP$^+$)-transduced HLCD19 KO cells were enriched in the cell culture, indicating that CD19 expression promotes Igβ signaling and provides a growth advantage for these cells (Fig 5F). Indeed, upon anti-Igβ stimulation, only the CD19, but not the EV-transduced control cells, displays calcium mobilization (Fig 5G). Upon B-cell activation, IgM-BCR and CD19 increase their proximity on the cell surface (Klasener *et al*, 2014). We used a Fab-based proximity ligation assay (Fab-PLA) to measure the proximity between CD19 and Igβ on WT and on HLα KO Ramos cells (Fig 6A). Interestingly, HLα KO Ramos cells display a stronger Igβ:CD19 PLA signal than WT Ramos B cells, even though the latter cells carry 50 times more Igβ on their cell surface (see Fig 2B). This analysis suggests that most of the expressed Igβ is localized in close proximity to the CD19 molecule on the HLα KO Ramos B-cell surface. The importance of Igβ for the fitness of Ramos B cells is also suggested by the fact that transcriptome profiling in these cells shows four-fold higher transcript levels for Igβ than Igα (Fig 6B). These data suggest that Igβ and CD19 are part of an ITAM and PI3K signaling module that promotes the survival of BL tumor cells (Fig 6C). In line with this proposal is the finding that the proliferation of WT and α KO Ramos cells can be inhibited by the PI3K inhibitor CAL-101 whereas the null and β KO Ramos cells that both have lost Igβ expression and signaling function are less sensitive to this inhibitor (Fig 6D).

To test whether or not the increased Igβ:CD19 proximity is a general feature of BCR-negative B cells, we generated Igα- and Igβ-deficient clones also from another BL cell line namely DG75. We found that in spite of their low Igβ expression, the Igα KO DG75 had a stronger Igβ:CD19 PLA signal than the WT DG75 B cells (Fig EV2). We next employed the B1-8f/Δ mb1CreERT2 mouse model to delete the B1-8VH gene segment in mature B cells, thus rendering these cells HC- and BCR-negative. We found that the BCR-negative B cells still displayed a strong Igβ:CD19 PLA signal (Fig EV3). The Igβ:CD19 signaling module thus is likely to be expressed on all BL as well as on normal B cells with reduced or defective BCR expression.

## Discussion

We here show that the competitive fitness of the BL cell Ramos depends on the expression of Igβ and CD19 molecules and proper ITAM/PI3K signaling. Furthermore, we show that, in the absence of any other BCR component (H, L, and Igα), Igβ comes to the B-cell surface where it is localized in close (10–20 nm) proximity to CD19. We propose that Igβ and CD19 are part of an alternative B-cell signaling module that promotes the survival of B lymphoma and also of normal B cells.

B cells co-express two BCR signaling components, Igα and Igβ, each of which carries an ITAM sequence with partially redundant functions (Rolli *et al*, 2002). It is thus surprising that only Igβ and not Igα can support the competitive fitness of Ramos B cells. This distinct behavior is apparently not due to a different signaling function of the cytosolic tail of Igα and Igβ. Indeed, the Igα tail can signal fitness in the context of an Igβ-α tail chimera and is even better than WT Igβ in supporting the growth of Ramos B cells. Rather, it seems to be the special features of Igβ that is responsible for this difference. Only Igβ, but not Igα, seems to be able to form a homodimer that is transported to the B-cell surface in the absence of any other BCR components (Radaev *et al*, 2010). Indeed, the exposure of Igβ-only Ramos B cells to anti-Igβ antibodies induces a calcium flux whereas Igα-only Ramos B cells did not respond at all to anti-Igα antibodies. This finding supports the notion that Igβ has a unique ability to be expressed in a mIg-independent manner and to form an alternative ITAM-signaling module on the B-cell surface. However, it has also been found that the Igα/Igβ heterodimer is expressed in an mIg-independent manner on the surface of pro-B cell where it may be part of a pro-B-cell specific signaling module (Nagata *et al*, 1997; Maki *et al*, 2000).

Tumor cell lines are "addicted" to certain oncogenes, the expression of which is required for their continuous growth in culture. For example, BL cells have a deregulated Myc and TCL1A expression, which promotes growth and survival, respectively (Dalla-Favera *et al*, 1982; Hoyer *et al*, 2002). Interestingly, Ramos B cells contain four times more transcripts of Igβ than Igα, whereas in normal B cells, Igα expression is equal to or stronger than Igβ expression. In the presence of Igα, the formation of an Igα/Igβ heterodimer may be more efficient than Igβ/Igβ homodimerisation, thus favoring assembly of the BCR complex. The prominent Igβ production in Ramos B cells may favor Igβ/Igβ homodimer production and Igβ/CD19 signaling. In this respect, Igβ may function as a tumor promoter and oncogene for BL cells.

In the context of the complete BCR, the ITAM sequences of the Igα/Igβ heterodimer seem to be protected by cytoskeletal elements in resting B cells and only become accessible to Syk upon B-cell activation and the opening of the BCR. As part of the Igβ/Igβ homodimer, however, the ITAM tyrosines may be less well protected and directly accessible to Syk. Furthermore, on the surface of resting B lymphocytes, the CD19 co-receptor resides, together with several other proteins, inside an IgD-class protein island. Only upon B-cell activation does CD19 change its location and can be found in close proximity to the open and active IgM-BCR (Klasener *et al*, 2014). Interestingly, we have previously observed that Syk-deficient B cells are critically dependent on CD19 expression (Hobeika *et al*, 2015). However, on the surface of Ramos B cells and prior to any stimulation, CD19 is already found in close vicinity to Igβ. The constitutive

Igβ and CD19 co-localization and the open, accessible ITAM sequence of Igβ seem be the two major features that are responsible for the continuous signaling behavior of the Igβ/CD19 receptor module.

We found that the formation of the Igβ/CD19 module is not only restricted to Ramos B cells but also expressed on other BL tumor lines and even on normal B cells with defective BCR expression. It is thus feasible that the Igβ/CD19 module also plays a role on normal B cells. For example, it has been found that after their activation, splenic B cells rapidly internalize their BCR but still can be stained with anti-Igβ antibodies although they are mIg negative (Kremyanskaya & Monroe, 2005). It was suggested that upon ligation, the BCR complex is destabilized so that the mIg molecule is separated from the Igα/Igβ heterodimer which stays alone on the B-cell surface. An alternative interpretation of these data is that the Igβ/CD19 module is also expressed on normal B cells and that it becomes detectable after BCR internalization. Continuous signaling from the Igβ/CD19 receptor module may also be the explanation for the long-term survival of mature murine B cells that upon inducible Cre-mediated deletion of the Igα gene, become BCR-negative (Levit-Zerdoun *et al*, 2016). Indeed, similar to what we have observed with Igβ-only Ramos B cells, Igα-deficient murine B cells can flux calcium upon exposure to anti-Igβ antibodies. Thus, low amounts of the Igβ/CD19 receptor module may also be expressed and detected on normal B cells under certain conditions (low Igα expression and BCR internalization) and play an important role for the survival of these cells. That only Igβ but not Igα has a special B-cell survival function is also suggested by a study in mice with an inducible deletion of either the Igβ or Igα tail (Song *et al*, 2016). Interestingly, the expression of Igβ is specifically down-regulated in germinal center B cells, suggesting that during the selection for high-affinity B-cell clones, the Igβ/CD19 signaling module is switched off (Todo *et al*, 2015).

Most B-cell tumors seem to be dependent on a continuous or tonic BCR signal for their proper growth (Kuppers, 2005). A recent study of murine BL cells showed, similar to our data, that BCR-negative cells are less fit in a competitive growth assay (Varano *et al*, 2017). However, in this study, it was the loss of the H chain that was responsible for the reduced fitness. It would be interesting to test whether these cells also require Igβ expression. Whether or not the Igβ/CD19 module also play a role on other B-cell tumors apart from BL is not clear right now. However, the tumor-promoting effect provided by CD19 and active ITAM signaling may also be reached by other means. For example, all human B-CLL cells carry an auto-aggregated autonomously signaling BCR (Duhren-von Minden *et al*, 2012). With our Fab-PLA technique, we found that these receptors not only have an open conformation, but are also in close proximity to the CD19 protein, thus combining PI3K signaling with active ITAM signaling (unpublished observation). In line with this, the growth of B-CLL cells is blocked by Syk or PI3K inhibitors (Gobessi *et al*, 2009; Danilov, 2013). The BCR is also stably expressed on all ABC-DLBCLs, although the growth of these tumors is driven more by active NFkB than by PI3K signaling (Davis *et al*, 2010). Interestingly, ABC-DLBCLs frequently carry a mutation of the first ITAM tyrosine of Igβ, which we now know to be required for the signaling function of the Igβ/CD19 module. Whether it is the block of the Igβ/CD19 module and/or of BCR internalization that allows a more stable BCR expression on these cells is not yet clear. It will therefore be important to learn more about the conformation and the immediate nano-environment of the BCR on the ABC-DLBCLs B cells.

Previously, it was only possible to generate mutations of mammalian genes by time-consuming and inefficient procedures such as homologous recombination in ES cells (Bouabe & Okkenhaug, 2013). With the advent of the CRISPR/Cas9 technology, any gene can be deleted efficiently and rapidly in many different cell lines, as long as the deletion is compatible with the survival of the cells (Hsu *et al*, 2014). This revolutionary technology is a game changer for the study of oncogenic signaling of human tumor cells that previously relied on knock-down and overexpression approaches. Importantly, with the CRISPR/Cas9 technology, it is feasible to sequentially delete several genes. This allows B-cell engineering approaches at a new level. In our study, we have deleted seven human genes in different combinations and demonstrate that this results in new insights into the transformation process and the regulation of tumor B-cell survival. In combination with the rapidly increasing genomic data on human tumors, and new techniques studying nanoscale organization of receptors and signaling components, the CRISPR/Cas9 technology will be a powerful tool for the better understanding and finally better treatment of human tumors.

## Materials and Methods

### Antibodies

For FACScan analysis and screening of different gene knockout (KO) cells, the following anti-human antibodies were used: anti-CD79A-PE (HM47, BioLegend), anti-CD79B-PE (CB3-1, BioLegend), anti-IgM-APC (G20-127, BD Biosciences), mouse IgG1, κ isotype control-APC (MOPC-21, BD Biosciences), anti-IgD-PE (IA6-2, BD Biosciences), mouse IgG2a, κ isotype control-PE (G155-178, BD Biosciences), anti-Igλ-APC (1-155-2, eBioscience), anti-CD19-Alexa fluor 647 (HIB19, BioLegend), and anti-CD81-APC (1D6, eBioscience). To stimulate cells and to measure calcium responses, anti-CD79B (CB3-1, BioLegend), anti-CD79B (EPR6860, Abcam), and anti-CD19-Alexa fluor 647 (HIB19, BioLegend) antibodies were used. To prepare PLA probes and F(ab')$_2$ fragments, the following antibodies were used: anti-human CD79B (CB3-1, Acris), anti-human CD19 (HIB19, Biolegend). The following antibodies were used for Western blot (WB): anti-IgM (Southern Biotech), anti-IgD (Southern Biotech), anti-lambda (Southern Biotech), anti-CD79A (HM47, BioLegend), anti-CD79B (CB3-1, Southern Biotech), anti-BLNK (2B11, Santa Cruz Biotechnology), and anti-GAPDH (6C5, Thermo Fisher).

### Cell culture

Ramos and DG75, the Epstein–Barr virus-negative human BL lines, were cultured in RPMI medium (Gibco) supplemented with 10% FCS (Biochrom), 10 units/ml penicillin/streptomycin (Gibco), and 50 mM β-mercaptoethanol (Sigma) at the 37°C with 5% $CO_2$. The Phoenix cell line was cultured in Iscove's medium (Biochrom) supplemented with 10% FCS (Biotech), 10 units/ml penicillin/streptomycin (Gibco), and 50 mM β-mercaptoethanol (Sigma) at 37°C 7.5% $CO_2$.

### Experimental mice

We crossed the floxed B1-8f/Δ Ig transgenic mice (Lam *et al*, 1997) with mb1-CreERT2 mice (Hug *et al*, 2014) to generate the B1-8f/Δ mb1CreERT2 mice (Hobeika *et al*, 2015; Levit-Zerdoun *et al*, 2016). The mice were induced as reported previously to generate BCR⁻ B cells (Becker *et al*, 2017). Splenic B cells were isolated 8 days after tamoxifen treatment. All animal studies were conducted in mice aged 10–14 weeks and were carried out at the Max Planck Institute of Immunobiology and Epigenetics animal facilities in accordance with the German Animal Welfare Act, having been reviewed and approved by the regional council.

### Flow cytometry (FACS) analysis

For surface staining, $1–20 \times 10^5$ cells were stained with antibodies in PBS on ice for 20 min and then measured with an LSR II (BD). For intracellular staining, the cells were processed with the ADG FIX & PERM Kit according to the manufacturer's protocol (Dianova). Data were exported in FCS-3.0 format and analyzed with FlowJo software (TreeStar). A FACSAria (Becton Dickinson) cell sorter was used to separate single cells or cell populations.

### CRISPR/Cas9 knockout

All CRISPR/Cas9 KO plasmids used in this study are listed in Appendix Table S1. CRISPR/Cas9 deletion was carried out using the Neon transfection system (Invitrogen) to deliver the KO plasmids into the cells. For one reaction, $1 \times 10^6$ cells were resuspended with 100 μl transfection medium containing 20 mM HEPES (Gibco) and 1.25% DMSO (Sigma) in RPMI medium and then mixed together with 4 μg of KO plasmid. The cells were then transfected using a single pulse at 1,350 V, with a 30 ms pulse-width. The transfected cells were then cultured at 37°C and 5% $CO_2$. The transfection efficiency was visualized by the transient expression of GFP in the cells 24h to 48h post-transfection by FACScan. The GFP⁺ viable cells were sorted into 96-well plates with one cell per well. The single sorted cells were cultured for 10–14 days before the individual colonies were transferred into larger wells for expansion. Inactivation of the target gene was verified by FACScan using antibody staining and/or WB and/or genotyping.

### Retroviral transduction

Retroviral transductions in Ramos cells were performed as previously described (Storch *et al*, 2007). In brief, Phoenix cells were transfected using PolyJet DNA *in vitro* transfection reagent following the manufacturer's protocol (SignaGen Laboratories). Retrovirus-containing supernatants were collected 48 h after transfection and used for transduction.

### Constructs

The sgRNAs targeting genes of mIg H chain, L chain, and CD19 were inserted into pSpCas9(BB)-2A-GFP (PX458, Addgene ID # 48138) following the Ran's protocol (Ran *et al*, 2013). The open reading frame of the Igβ and CD19 was cloned from Ramos cDNA and inserted into the pJET vector by CloneJET PCR Cloning Kit

(Thermo Scientific). The retroviral constructs were created with the In-Fusion® HD Cloning Kit (Clontech) using pMIG-IRES-GFP vector as the backbone. The ITAM mutants of Igβ were created by QuickChange Site-Directed Mutagenesis Kit (Agilent Technologies). Primers for all cloning are listed in Appendix Table S2. All of the inserts were verified by sequencing (MPI-IE, Freiburg, Germany).

### Cell proliferation assay

The cell proliferation assay was performed using CytoTell™ Ultra-Green (AAT Bioquest) according to the manufacturer's protocol.

### Cell growth competition assay

For the growth competition assay, KO cells were co-cultured with GFP-expressing wild-type cells (gWT). All KO cell lines were originally derived from a single sorted cell. The gWT cells were produced by retrovirally transducing WT Ramos with pMIG resulting in GFP expression in transduced cells. gWT cells were used for all competition assays with cells carrying mutations of the BCR components. For the competition assay with HLSLP65 KO cells, the GFP-tagged HL-KO cells were used as a competitor. For the assay, KO (GFP⁻) and competitor (GFP⁺) ($1 \times 10^5$ cells each) were mixed and co-cultured in one well of a 12-well plate. An aliquot of the mixture was sampled, and new medium was refilled each day from day 1 to day 8. The relative growth of the WT and KO cells was estimated using the ratio of the GFP⁻ and GFP⁺ cell populations as determined by FACScan.

### Calcium measurement

Calcium measurements were performed as previously described (Storch *et al*, 2007). In brief, $1 \times 10^6$ cells were loaded with indo-1 (Molecular Probes) following the manufacturer's instructions and resuspended in 500 μl Iscove's medium supplemented with 1% FCS. Before analysis, the cells were pre-warmed at 37°C for 5 min. Stimuli were added after 1 min of baseline recording. The calcium flux was measured with a Fortessa II (BD). For kinase inhibition in the calcium measurements, indo-1-loaded cells were pre-treated with 5 μM Syk inhibitor R406 (Selleckchem, Houston, TX) or 10 μM SRC-family kinases (SFK) inhibitor PP2 (Sigma-Aldrich) for 5 min at 37°C. Data were exported in FCS-3.0 format and analyzed with FlowJo software (TreeStar).

### PLA probe preparation and experiment

For Fab-PLA, F(ab')₂ fragments were prepared from the corresponding antibodies using the Pierce Fab Micro preparation Kit (Thermo Fisher Scientific) according to the manufacturer's protocol. In brief, F(ab')₂ fragments were desalted (Zeba™ spin desalting columns, Thermo Fisher Scientific) and coupled with PLA Probemaker Plus or Minus oligonucleotides according to the manufacturer's protocol (Sigma-Aldrich) to generate Fab-PLA probes. For *in situ* PLA experiments, the cells were allowed to attach to polytetrafluoroethylene (PTFE)-coated slides (Thermo Fisher Scientific) for 30 min at 37°C. Depending on the experiment, the cells were activated or treated with inhibitors and then fixed for 20 min with 4% paraformaldehyde in PBS. PLA was performed as previously described (Klasener

*et al*, 2014). In brief, after incubation with a blocking solution containing 25 μg/ml sonicated salmon sperm DNA and 250 μg/ml bovine serum albumin (BSA) in PBS, the cells were incubated with Fab-PLA probes in PBS. PLA signal amplification was performed following the manufacturer's protocol. The resulting samples were directly mounted on slides with DAPI-Fluoromount-G (Southern Biotech) to visualize the PLA signals in relation to the nucleus.

### Imaging and image analysis

All microscope images were acquired using a Zeiss 780 Meta confocal microscope (Carl Zeiss) equipped with a Zeiss Plan-Apochromat 63× oil immersion objective lens. For each sample, several images were captured from randomly chosen regions. All recorded images were analyzed with BlobFinder in parallel to ImageJ software. PLA signals (dots/cells) were counted from at least 500 cells for each sample.

### Data processing and statistical analysis

Raw data produced by BlobFinder/ImageJ were exported to Prism software (GraphPad, La Jolla, CA). The mean PLA signal count per cell was calculated from the corresponding images, and the statistical significance was calculated with nonparametric Mann–Whitney test.

### Western blot

Cells were collected and immediately lysed on ice in lysis buffer containing 1% Triton X-100. Cleared lysates were subjected to 10% SDS–PAGE and subsequent immunoblotting.

### Treatment with PI3K inhibitor

The treatment with PI3K inhibitor CAL-101/idelalisib (5–20 μM; Selleckchem) was performed as described in a recent study of Varano *et al* (2017).

### Phospho flow cytometry

HL-KO and HLβ KO cells were activated with 5 mM PV for 60 s. The cells were fixed and stained according to the Biolegend "Intracellular Staining With True-Phos™ Perm Buffer in Cell Suspensions Protocol" for pSyk Y352 (PE-labeled, Biolegend 683703) and Syk (APC-labeled, Biolegend 644305) or the appropriate isotype controls and then measured with an Attune NXT (Thermo Fisher).

**Expanded View** for this article is available online.

### Acknowledgements

We thank Dr. Lise Leclercq for critical reading of this manuscript. We thank Prof. Dr. Hassan Jumaa for the tamoxifen-inducible SLP65ERT2 plasmid. We thank Prof. Dr. Klaus Rajewsky for the floxed B1-8f/Δ Ig transgenic mice. This study was supported by the German Cancer Foundation grant 111026, by the Deutsche Forschungsgemeinschaft through TRR130-P02 and SFB746-P07 and by the Max Planck Society. Support came also through the ERC-grant 322972 to M.R. and the Excellence Initiative of the German Federal and State Governments (EXC 294).

### Author contributions

XH, JY, and MR planed the experiments. Most experiments were conducted by XH; KK conducted the PLA study in Ramos and DG75 cells; MB and PCM conducted the PLA study in B1-8f/Δ mb1CreERT2 mice; MC conducted the phosphor flow experiment; JMI and MB provided cell lines and targeting vectors, and PJN helped with statistic and bioinformatics. The manuscript was written by XH, JY, and MR.

### Conflict of interest

The authors declare that they have no conflict of interest.

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
