## [Review Process File · The EMBO Journal]

Continuous signaling of CD79b and CD19 is required for the fitness of Burkitt lymphoma B cells

Xiaocui He, Kathrin Kläsener, Joseena M. Iype, Martin Becker, Palash C. Maity, Marco Cavallari, Peter J. Nielsen, Jianying Yang and Michael Reth.

Review timeline:

Submission date:	11 th August 2017
Editorial Decision:	18 th September 2017
Revision received:	8 th February 2018
Editorial Decision:	21 st February 2018
Revision received:	28 th February 2018
Accepted:	7 th March 2018

Editor: Karin Dumstrei

Transaction Report:

1st Editorial Decision

18th September 2017

Thank you for submitting your manuscript to The EMBO Journal. Your study has now been seen by three referees and their comments are provided below.

As you can see from the comments, the referees find the analysis interesting. However, it is also clear that the study has to be extended in order to consider publication here. In particular the referees bring up the point that the findings need to be extended to another cell lines. Given the referee reports, I would like to invite you to submit a revised manuscript should you be able to address the concerns raised in full. I should add that it is EMBO Journal policy to allow only a single major round of revision and that it is therefore important to address them at this stage.

Thank you for the opportunity to consider your work for publication. I look forward to your revision.

REFeree REPORTS.

Referee #1:

The manuscript from He and colleagues identifies Ig β likely in association with CD19 as an important cell surface molecule that promotes the survival of Ramos Burkitt lymphoma. Indeed, using CRISPR/Cas9 technology to delete various BCR components, they show that Ig β but not Iga can activate the PI3K signaling pathways, which allows the proliferation of RAMOS cell line in the absence of BCR expression. In addition, the authors show that CD19 plays an important role in the induction of Ig β -induced calcium flux and appears to form a complex with Ig β on the cell surface of Ramos cells as measured by Fab-based proximity ligation assay. Altogether, the manuscript is well written and the provided data support the conclusions of the authors.

Although CD19/Ig β complexes may not be easily identifiable in non-transformed B cells, it would be important that this association could be detected in other Burkitt B cell lines so that this observation may not appear to be restricted to Ramos B cells.

In addition, the association of Ig β with CD19 should be further evidenced in co-immunoprecipitation experiments and western blots. This approach may also identify additional molecules that may participate to CD19/Ig β complex formation. For instance, is Calnexin that binds Ig α /Ig β complexes at the pro-B cell stage and in the absence of BCRs (Nagata et al. *Immunity* 1997) also associated with CD19/Ig β in RAMOS B cells?

Finally, the discussion should include references from the Karasuyama group that reported efficient Ig β signaling in Rag2-deficient pro-B cells after antibody crosslinking and that involves similar events described in the current manuscript (Nagata et al. *Immunity* 1997 and Maki et al. *JEM* 2000).

Minor points

- 1- Wild type controls should be included in Figure 4C and 5C.
- 2- symbol font for Ig β was not displayed properly in Figures 5 and 6.

Referee #2:

In this report, He and colleagues provide a comprehensive analysis of the effects of genetic inactivation of one or more components of the B cell antigen receptor complex on the competitive growth properties of the human Burkitt lymphoma cell line, RAMOS.

In order to generate gene knock-outs of BCR components in lymphoma cells, authors take advantage of the CRISPR/CAS9 technology. Specifically, they apply a transfection protocol to introduce (transiently?) into malignant B cells a Cas9 expression cassette in combination with gRNAs targeting the genes of interest. Single cell sorting of tumour cells subjected to the gene editing technology followed by their in vitro clonal expansion led to the isolation of lymphoma B cell derivatives lacking expression of the proteins of interest. A strategy based on sequential gene editing ensured the generation of compound mutant lymphoma lines lacking up to 4 genes. The net result of these gene manipulation was the creation of RAMOS derivatives lacking the immunoglobulin heavy and light chains (Ig HL) alone or in combination with each of the two BCR signalling components, Ig α and Ig β or with both. Mutant lymphoma cells were monitored for their in vitro growth properties once co-cultured with control RAMOS cells that were proficient for BCR expression and signalling.

Main findings

Through the monitoring of in vitro competition assays, authors report:

- Unperturbed lymphoma fitness upon concomitant extinction of IgH and IgL chain expression. The same result was achieved analysing lymphoma cells lacking the BCR signalling component Ig α ;
- Reduced fitness of RAMOS cells lacking Ig β alone or in combination with either Ig α or IgH and IgL chains;
- Residual Ig β expression in RAMOS cells lacking the immunoglobulin receptor (HL KO) alone or in combination with Ig α (HL α KO);
- Signalling competence of residual Ig β expressed on the surface of surface Ig-less lymphoma cells, as revealed by the transient increase in intracellular calcium levels in response to antibody-based anti-Ig β crosslinking;
- Dependency on both Ig β ITAM tyrosines and on the proximal BCR signalling effectors Syk and SLP65 for Ig β induced calcium signalling in RAMOS cells lacking Ig expression upon stimulation with an anti-Ig β antibody;

Together these results support a scenario whereby in the absence of sIg expression, membrane-bound Ig β but not Ig α sustains the fitness in vitro of BCR-less Burkitt lymphoma RAMOS cells.

Authors next hypothesized that residual Ig β expressed in Ig-less tumour cells contributed to lymphoma fitness forming a complex with the BCR co-receptor complex CD19/CD81. To address this point, He et al. generated RAMOS derivatives in which the lack of Ig HL chains was combined to deficiency in either CD19 or CD81. These experiments revealed that:

- Differently from HL mutants, HLCD19KO (and HLCD81KO) triple KOs disappeared in culture over time
- CD19 deficiency blunted intracellular calcium release in sIg-negative RAMOS cells triggered with an anti-Ig β antibody
- Reconstitution of HLCD19KO lymphoma cells with a CD19 expression vector restored fitness and Ig β signalling proficiency.

Interestingly, using a proximity ligation assay (PLA), Ye and colleagues report that despite the reduced Ig β levels measured by flow cytometry in Ig-less (HL α KO) RAMOS cells, there was a substantial increase in the mutant cells of CD19/Ig β PLA signals, when compared to wild-type cells. These results, may hint to a compensatory mechanism selected by Ig-less RAMOS cells in order to sustain their fitness, which is centred on the increased formation of CD19/Ig β complexes, possibly facilitated by the contemporary loss of BCR expression. The signalling competence of such complexes is suggested by the resistance of Ig β -null RAMOS cells to pharmacological inhibition of the CD19 downstream effector PI3K δ .

In summary, using a straight-forward gene knock-out approach, He and colleagues provide convincing evidence that in the MYC-transformed RAMOS tumour line, Ig β is critical to sustain the fitness of malignant B cells that have lost Ig expression. This role is exerted interacting functionally with the BCR co-receptor CD19.

Despite a number of caveats and open questions that remain to be addressed (outlined below), this study extends our current understanding of the possible mechanisms through which the BCR signalling complex influences lymphoma fitness. Revealing whether the described observations are unique to RAMOS cells or reflect a behavior common to at least a subset of Burkitt lymphomas and possibly other B cell malignancies becomes a needed, attractive and clinically relevant area of investigation. The study by He and colleagues also opens new perspectives on the possible contribution of BCR-independent Ig β modulation of normal mature B cell survival/fitness, especially when the antigen receptor is temporarily lost, or strongly down-regulated, such as in defined stages of the germinal centre reaction.

Main criticisms

- The experimental design by He and colleagues is based on (transient?) transfection of CRISPR/Cas9 vectors followed by single cell sorting and extended in vitro culture to ensure expansion of the subclones. This approach entails the intrinsic risk of selecting variants which have acquired the capacity to overcome the effects of the genetic mutation that was introduced. In this context, the ability of Ig β to form a preferential complex with CD19 in cells that have lost sIg expression could represent a compensatory mechanism selected by RAMOS cells to overcome the lack of a fitness signal provided otherwise by a conventional BCR complex. This possibility is supported by evidences shown in Fig 6A, indicating that despite the significant reduction of total Ig β levels seen in HLKO cells, (see Sup Figure 1D), the number of Ig β /CD19 PLA signals observed in these cells is greatly increased in comparison to wild-type cells. To prove that Ig (HLKO) expression in RAMOS is not required for tumour fitness, authors should monitor the fitness of Ig less cells soon after gene inactivation (i.e. in the pool of Cas9 gene edited cells few days after induction of the KO). These data are not provided. Therefore, whereas the current data support a model whereby at least in RAMOS cells, Ig β together with CD19 promotes fitness of the tumour cells that have lost Ig expression, they lack information of whether this function is conserved in malignant cells that retain a functional BCR on the surface. Authors are recommended to address and properly discuss this point.
- The current manuscript lacks information on the number of independent clones that were analysed for each type of BCR mutation. Indeed, the genetic instability featured by Burkitt lymphoma cells such as RAMOS may lead to the selection during the course of the time-consuming single cell cloning experiment, of independent variants, which may differently impact on the fitness of the tumour cells. Authors are recommended to provide fitness data for at least two independent subclones for each genetic mutation that was investigated.

- The calcium signalling data obtained in HLKO cells subjected to anti-Ig β crosslinking provide convincing evidence that Ig β is able to deliver signals in Ig-less RAMOS lymphoma cells and that this requires expression of a functional CD19 receptor. However, whether a spontaneous Ig β /CD19 concerted signalling cascade involving activation respectively of Syk, SLP65 (via Ig β) and PI3K δ (via CD19) is actively operating in unstimulated Ig-negative RAMOS cells to sustain tumour cell fitness remains to be demonstrated. Comparing p-SYK, p-SLP65 and/or pAKT levels between HLKO and HL β KO cells grown in isolation or retrieved from competitions may help address this point. Also, could PLA be applied to detect the presence of Ig β /pSyk nanocomplexes in HLKO cells?
- The present manuscript lacks any information on the growth and survival properties of RAMOS derivatives losing one or more components of the BCR complex. Growth curve analysis and survival properties of wild-type and tumour cells measured in isolation and during competitions will help understanding how Ig β (but not Ig α) supports the competitive growth of Ig-less RAMOS cells.
- Ig β levels: data shown in Sup. Figure 1E suggest (although quantification of the results was not provided) that the total pool of Ig β molecules increases in RAMOS cells upon HL and/or Iga inactivation. This result contrasts with flow-cytometric data showing a reduction of Ig β protein levels in HL KO cells (Figure 2B). Authors should clarify this point providing quantitative measurements by immunoblotting analysis of Ig β levels respectively in wild-type, HLKO, HL α KO and α KO. Analysis of Ig β transcripts in the same lymphoma subsets should complete the analysis. This point will help clarify whether chronic loss of HL induces/selects for changes in Ig β expression, or whether the small amount of residual Ig β present in the cell gets fully recruited to CD19 to sustain fitness of Ig-less tumour B cells.
- Rescue of HL β KO fitness by reconstitution with Ig β tail constructs: the failure of Ig β tail mutants to rescue HL β KO cells could be explained by the lower expression of the retrovirally-encoded proteins in the cells, when compared to those coding for wt Ig β or Ig β α -tail (see Figure 3A). To exclude this possibility, authors should provide quantification data (by immunoblotting or showing MFI data obtained through flow-cytometry) of the expression levels of the various constructs introduced into HL β KO cells.
- A clear limitation of this study is the confinement of the results to a single tumour cell line. Extending the main results to at least a second tumour line would help support the author's conclusion. Screening by PLA the existence of other lymphoma lines possibly displaying spontaneous Ig β /CD19 nanocomplexes (such as those shown in Figure 6A) could help select those that, like RAMOS, depend on the Ig β /CD19 complex for optimal fitness once BCR expression is lost.

Minor points

- Figure 2A, 3A: provide MFI data for each mutant/complemented tumor population to better appreciate the expression levels of the corresponding proteins.

Referee #3:

Continuous signaling of CD79b and CD19 are required for the fitness of Burkitt lymphoma B cells

It is well established that BCR expression is essential for development as well as for the maintenance of mature B cells. Burkitt lymphoma requires continuous BCR signaling for their tumor growth. This is driven by ITAM and PI3K signaling.

The authors show, using CRISPER/Cas9 technologies to delete BCR as well as co-receptor genes in human BL cell line Ramos, that the competitive fitness of the BL cell line depends on the expression of Ig beta and CD19 and proper ITAM signaling.

The authors propose that Ig beta and CD19 are part of an alternative B cell signaling module that promotes the survival of BL cells and also normal B cells.

Further they show in this paper that in the absence of any BCR component Ig beta can be expressed on the surface close to CD19 and signals in an ITAM dependent manner.

With this data they claim that Ig beta and CD19 are part of an alternative B cell signaling module that use continuous ITAM/PI3K signaling to promote the survival of B cell lymphoma and normal B cells.

Comments:

The paper is well written and the experiments are of high quality and underline the presented theory.

Main criticism:

The study relies completely on one cell line type and this cell line is derived from a cancer patient, namely these are not normal B cells. Although the results are very interesting, one remains wondering if this is the case also for "normal" B cells, as suggested by the authors.

There are many mouse lines with mutations in the BCR components, as well reviewed in the introduction by the authors.

I would like the authors to repeat, at least their main findings with cells from mouse mutants that are similar to the mutations shown in this paper.

One possible source of BCR deficient cells could be the system recently published by the groups of Rajewsky and Casola in Nature, where the mice express MYC and lack BCR expression. Although this system also makes use of malignant B cells, it would be interesting to use it to study whether in other systems Ig beta is expressed on the surface of the cells in the absence of the BCR.

Minor comments:

1. The authors should show their demonstrating that Ramos B cells lacking Ig alpha cannot respond by calcium flux to anti-Ig alpha, as they did for the Ig beta.
2. Fig. 3H, make sure the arrow is pointed correctly.
3. Please quantify the levels of Ig beta in 3A and 3E, its hard to see differences when data is presented in these histograms.
4. It might be a problem of mac/PC, but on my computer some symbols appear as unknown signs. For example in the text of page 7. Please check.

1st Revision - authors' response

8th February 2018

Referee #1:

The manuscript from He and colleagues identifies Ig β likely in association with CD19 as an important cell surface molecule that promotes the survival of Ramos Burkitt lymphoma. Indeed, using CRISPR/Cas9 technology to delete various BCR components, they show that Ig β but not Ig α can activate the PI3K signaling pathways, which allows the proliferation of RAMOS cell line in the absence of BCR expression. In addition, the authors show that CD19 plays an important role in the induction of Ig β -induced calcium flux and appears to form a complex with Ig β on the cell surface of Ramos cells as measured by Fab-based proximity ligation assay. Altogether, the manuscript is well written and the provided data support the conclusions of the authors.

We thank the reviewer 1 for this positive judgment.

Although CD19/Ig β complexes may not be easily identifiable in non-transformed B cells, it would be important that this association could be detected in other Burkitt

B cell lines so that this observation may not appear to be restricted to Ramos B cells.

*We agree with the reviewer 1 that it is important to show that the BCR independent expression of Ig β and the close proximity between CD19 and Ig β on the B cell surface is not only a feature of Ramos B cells. We thus have generated Ig β or Ig α KO mutants from another IgM-BCR carrying human Burkitt lymphoma line namely DG75 and analyzed with Fab-PLA the CD19 and Ig β proximity on these mutants and on DG75 WT B cells. The **new Fig. EV2** shows that the Ig α KO mutant display in comparison to DG75 WT B cells an increased CD19 and Ig β Fab-PLA signal although the KO cells carry less Ig β protein on their cell surface than the DG75 WT B cells. Thus DG75 behave identical to Ramos B cells in this respect. Furthermore, we show that normal murine spleen cells that loose their H-Chain and BCR expression after a tamoxifen induced Cre-mediated VH exon deletion still maintain Ig β in close proximity to CD19 on their cell surface as indicated by the strong Ig β /CD19 Fab-PLA signal (**new Fig. EV3**). This is in line with our study (**Levit-Zerdoun et al. 2016**) of B cells from the inducible Ig α -deleter mouse that we discuss in our manuscript. Although the Ig α negative splenic B cells loose their BCR expression they still carry Ig β on their cell surface and can display a calcium response upon exposure to anti-Ig β antibodies. Thus the functional co-localisation of Ig β and CD19 is not restricted to Ramos B cells but seem to be a general feature of human BL tumor and murine splenic B cells.*

In addition, the association of Ig β with CD19 should be further evidenced in co-immunoprecipitation experiments and western blots. This approach may also identify additional molecules that may participate to CD19/Ig β complex formation. For instance, is Calnexin that binds Iga/Ig β complexes at the pro-B cell stage and in the absence of BCRs (Nagata et al. Immunity 1997) also associated with CD19/Ig β RAMOS B cells?

We conducted several co-immunoprecipitation with anti-Ig β and copurified only in some cases CD19 in variable amounts with Ig β . We thus do not think that the two proteins form a stable Ig β /CD19 complex similar to CD19/CD81 and we thus avoid this word in our manuscript. What we think is that CD19 and Ig β are colocalized together inside a functional nano-compartment similar to what we recently described for the IgD-BCR and CXCR4 interaction (Becker et al. 2017). In both these case we show with Fab-PLA the close proximity of the two components as well as provid evidence for their functional connection. Calnexin seem not to play a role for the expression of Ig β or Ig α on Ramos B cells as in a cytometry analysis we could not detect it in on the surface of these cells (data not shown).

Finally, the discussion should include references from the Karasuyama group that reported efficient Ig β signaling in Rag2-deficient pro-B cells after antibody crosslinking and that involves similar events described in the current manuscript (Nagata et al. Immunity 1997 and Maki et al. JEM 2000).

We now mention the role of Ig β signaling for the induction of pre-B cell development in our discussion.

Minor points

1- Wild type controls should be included in Figure 4C and 5C.

The competition growth assay shown in Fig. 4C was conducted by mixing the HLSP65 KO with HL KO(GFP) Ramos B cells and showed that the HLSP65 KO cells are competed out by the HL KO cells. We control the experiment using the mixture of HL KO with HL KO(GFP). As we are dealing here with triple versus double KO cells we think that the HL KO Ramos is a better control than WT Ramos cells.

The experiment shown in Fig. 5C is actually not a mixing experiment. Rather we here used the CRISPR/Cas9 method to delete either the CD19 or the CD81 gene in the HL double KO Ramos B cell population and monitored the loss of the triple KO cell over time by a FACScan analysis. This experimental design does not allow adding WT Ramos cells as a control. However, we want to point out that in the competition growth assay shown in Fig. 2 HL KO cells perform as good as the WT Ramos cells.

2- symbol font for Ig β was not displayed properly in Figures 5 and 6.

We have changed the font in the new version of our manuscript.

Referee #2:

In this report, He and colleagues provide a comprehensive analysis of the effects of genetic inactivation of one or more components of the B cell antigen receptor complex on the competitive growth properties of the human Burkitt lymphoma cell line, RAMOS.

In order to generate gene knock-outs of BCR components in lymphoma cells, authors take advantage of the CRISPR/CAS9 technology. Specifically, they apply a transfection protocol to introduce (transiently? *yes*) into malignant B cells a Cas9 expression cassette in combination with gRNAs targeting the genes of interest. Single cell sorting of tumour cells subjected to the gene editing technology followed by their in vitro clonal expansion led to the isolation of lymphoma B cell derivatives lacking expression of the proteins of interest. A strategy based on sequential gene editing ensured the generation of compound mutant lymphoma lines lacking up to 4 genes. The net result of these gene manipulation was the creation of RAMOS derivatives lacking the immunoglobulin heavy and light chains (Ig HL) alone or in combination with each of the two BCR signalling components, Ig α and Ig β or with both. Mutant lymphoma cells were monitored for their in vitro growth properties once co-cultured with control RAMOS cells that were proficient for BCR expression and signalling.

Main findings

Through the monitoring of in vitro competition assays, authors report:

- Unperturbed lymphoma fitness upon concomitant extinction of IgH and IgL chain

expression. The same result was achieved analysing lymphoma cells lacking the BCR signalling component Ig α ;

- Reduced fitness of RAMOS cells lacking Ig β alone or in combination with either Ig α or IgH and IgL chains;
- Residual Ig β expression in RAMOS cells lacking the immunoglobulin receptor (HL KO) alone or in combination with Ig α (HL α KO);
- Signalling competence of residual Ig β expressed on the surface of surface Ig-less lymphoma cells, as revealed by the transient increase in intracellular calcium levels in response to antibody-based anti-Ig β crosslinking;
- Dependency on both Ig β ITAM tyrosines and on the proximal BCR signalling effectors Syk and SLP65 for Ig β induced calcium signalling in RAMOS cells lacking Ig expression upon stimulation with an anti-Ig β antibody;

Together these results support a scenario whereby in the absence of sIg expression, membrane-bound Ig β but not Ig α sustains the fitness in vitro of BCR-less Burkitt lymphoma RAMOS cells.

Authors next hypothesized that residual Ig β expressed in Ig-less tumour cells contributed to lymphoma fitness forming a complex with the BCR co-receptor complex CD19/CD81. To address this point, He et al. generated RAMOS derivatives in which the lack of Ig HL chains was combined to deficiency in either CD19 or CD81. These experiments revealed that:

- Differently from HL mutants, HLCD19KO (and HLCD81KO) triple KOs disappeared in culture over time
- CD19 deficiency blunted intracellular calcium release in sIg-negative RAMOS cells triggered with an anti-Ig β antibody
- Reconstitution of HLCD19KO lymphoma cells with a CD19 expression vector restored fitness and Ig β signalling proficiency.

Interestingly, using a proximity ligation assay (PLA), Ye and colleagues report that despite the reduced Ig β levels measured by flow cytometry in Ig-less (HL α KO) RAMOS cells, there was a substantial increase in the mutant cells of CD19/Ig β PLA signals, when compared to wild-type cells. These results, may hint to a compensatory mechanism selected by Ig-less RAMOS cells in order to sustain their fitness, which is centred on the increased formation of CD19/Ig β complexes, possibly facilitated by the contemporary loss of BCR expression. The signalling competence of such complexes is suggested by the resistance of Ig β -null RAMOS cells to pharmacological inhibition of the CD19 downstream effector PI3K δ . In summary, using a straight-forward gene knock-out approach, He and colleagues provide convincing evidence that in the MYC-transformed RAMOS tumour line, Ig β is critical to sustain the fitness of malignant B cells that have lost Ig expression. This role is exerted interacting functionally with the BCR co-receptor CD19.

Despite a number of caveats and open questions that remain to be addressed

(outlined below), this study extends our current understanding of the possible mechanisms through which the BCR signalling complex influences lymphoma fitness. Revealing whether the described observations are unique to RAMOS cells or reflect a behavior common to at least a subset of Burkitt lymphomas and possibly other B cell malignancies becomes a needed, attractive and clinically relevant area of investigation. The study by He and colleagues also opens new perspectives on the possible contribution of BCR-independent Ig β modulation of normal mature B cell survival/fitness, especially when the antigen receptor is temporarily lost, or strongly down-regulated, such as in defined stages of the germinal centre reaction.

We thank the reviewer 2 for his positive opinion on the implications of or findings.

Main criticisms

- The experimental design by He and colleagues is based on (transient?) transfection of CRISPR/CAs9 vectors followed by single cell sorting and extended in vitro culture to ensure expansion of the subclones. This approach entails the intrinsic risk of selecting variants which have acquired the capacity to overcome the effects of the genetic mutation that was introduced. In this context, the ability of Ig β to form a preferential complex with CD19 in cells that have lost sIg expression could represent a compensatory mechanism selected by RAMOS cells to overcome the lack of a fitness signal provided otherwise by a conventional BCR complex. This possibility is supported by evidences shown in Fig 6A, indicating that despite the significant reduction of total Ig β levels seen in HLKO cells, (see Sup Figure 1D), the number of Ig β /CD19 PLA signals observed in these cells is greatly increased in comparison to wild-type cells. To prove that Ig (HLKO) expression in RAMOS is not required for tumour fitness, authors should monitor the fitness of Ig less cells soon after gene inactivation (i.e, in the pool of Cas9 gene edited cells few days after induction of the KO). These data are not provided.

*We agree with the reviewer 2 that in KO experiments with cell lines (but actually also with mouse mutants) there is always the danger that one is selecting variants that are altered not only by the gene that one has targeted. To minimize this risk we have conducted our experiments several times and derived Ig β or Ig α deficient Ramos B cells by different routes (see **Fig. 1** and **Fig.2**). However, as suggested by reviewer 2, we also have conducted a new targeting experiment and analyzed the competitive growth of newly derived BCR negative Ramos clones only 14 days after the CRISPR/Cas9 mediated gene deletion (see **new Appendix Fig. S3**). In this experimental setting we again found that the Ig β deficient Ramos B cells are more rapidly lost from the culture than other BCR-negative Ramos B cells.*

Therefore, whereas the current data support a model whereby at least in RAMOS cells, Ig β together with CD19 promotes fitness of the tumour cells that have lost Ig expression, they lack information of whether this function is conserved in malignant cells that retain a functional BCR on the surface. Authors are recommended to address and properly discuss this point.

Unfortunately, it is currently experimentally not possible to detect the CD19 and Ig β containing nano-compartment (as mentioned above we do not think that CD19

*and Ig β form a defined protein:protein complex) on Ramos B cells that retain a functional BCR. All existing methods cannot distinguish between BCR associated and “free” Ig β on the B cell surface. The increased Ig β and CD19 proximity is however not only detected on BCR negative Ramos B cells but also on the human BL line DG75 and murine spleen cells once they become BCR negative (see **new Fig. EV2** and **Fig. EV3**). We think that our finding that human as well as murine B cells are able to form a CD19 and Ig β containing nano-compartment suggest that these structures have an evolutionary conserved function also on normal B cells and we will mention this in the discussion of our manuscript.*

- The current manuscript lacks information on the number of independent clones that were analysed for each type of BCR mutation. Indeed, the genetic instability featured by Burkitt lymphoma cells such as RAMOS may lead to the selection during the course of the time-consuming single cell cloning experiment, of independent variants, which may differently impact on the fitness of the tumour cells. Authors are recommended to provide fitness data for at least two independent subclones for each genetic mutation that was investigated.

*We think we are addressing this point with the explanation above and with the **new Appendix Fig. S3**. However we want to point out that we not only worked with isolated clones of Ramos cell. The CD19 and CD81 KO were batch sorted and thus are KO. We now explain this more explicit in the new version of our manuscript.*

- The calcium signalling data obtained in HLKO cells subjected to anti-Ig β crosslinking provide convincing evidence that Ig β is able to deliver signals in Ig-less RAMOS lymphoma cells and that this requires expression of a functional CD19 receptor. However, whether a spontaneous Ig β /CD19 concerted signalling cascade involving activation respectively of Syk, SLP65 (via Ig β) and PI3K δ (via CD19) is actively operating in unstimulated Ig-negative RAMOS cells to sustain tumour cell fitness remains to be demonstrated. Comparing p-SYK, p-SLP65 and/or pAKT levels between HLKO and HL β KO cells grown in isolation or retrieved from competitions may help address this point. Also, could PLA be applied to detect the presence of Ig β /pSyk nanocomplexes in HLKO cells?

*As Ig β is only expressed in rather low amounts on BCR-negative Ramos B cells it is unfortunately not possible to see phosphorylation event in unstimulated B cells. Only after exposure of HL KO and HL β KO Ramos B cells to pervandate did we detect by intracellular cytometry a reduced pSyk level in the latter Ramos B cells suggesting that the CD19/ Ig β nano-compartment contribute to Syk activation (see **new Fig. EV1**). Please note that Ramos B cell express, apart from Ig α /Ig β , also other ITAM containing molecules that are likely to be responsible for the residual Syk activation in the Ig β -deficient Ramos cells.*

- The present manuscript lacks any information on the growth and survival properties of RAMOS derivatives losing one or more components of the BCR complex. Growth curve analysis and survival properties of wild-type and tumour cells measured in isolation and during competitions will help understanding how Ig β (but not Ig α) supports the competitive growth of Ig-less RAMOS cells.

In the new Appendix Fig. S2, we now show the growth properties of several single (α and β) and double (α,β and HL) KO Ramos cells and found them similar to the Ramos WT and Ramos-null controls.

- Ig β levels: data shown in Sup. Figure 1E suggest (although quantification of the results was not provided) that the total pool of Ig β molecules increases in RAMOS cells upon HL and/or Ig α inactivation. This result contrasts with flow-cytometric data showing a reduction of Ig β protein levels in HL KO cells (Figure 2B). Authors should clarify this point providing quantitative measurements by immunoblotting analysis of Ig β levels respectively in wild-type, HLKO, HL α KO and α KO. Analysis of Ig β transcripts in the same lymphoma subsets should complete the analysis. This point will help clarify whether chronic loss of HL induces/selects for changes in Ig β expression, or whether the small amount of residual Ig β present in the cell gets fully recruited to CD19 to sustain fitness of Ig-less tumour B cells.

On a closed look at the new Appendix Fig. S1 E we do not think that in comparison to the GAPDH control the Ig β levels are altered in the different still Ig β -producing Ramos cells. The main purpose of Fig. S1E is to verify the loss of protein production in the different analyzed Ramos BCR-KO cells.

- Rescue of HL β KO fitness by reconstitution with Ig β tail constructs: the failure of Ig β tail mutants to rescue HL β KO cells could be explained by the lower expression of the retrovirally-encoded proteins in the cells, when compared to those coding for wt Ig β or Ig β α -tail (see Figure 3A). To exclude this possibility, authors should provide quantification data (by immunoblotting or showing MFI data obtained through flow-cytometry) of the expression levels of the various constructs introduced into HL β KO cells.

As suggested by reviewer 2 we now show the MFI data in the modified Fig.2 and Fig.3. Please note that at least on α,β KO Ramos the Ig β -tl does not show a drastic expression difference in comparison to Ig β -WT or Ig β - α tl.

- A clear limitation of this study is the confinement of the results to a single tumour cell line. Extending the main results to at least a second tumour line would help support the author's conclusion. Screening by PLA the existence of other lymphoma lines possibly displaying spontaneous Ig β /CD19 nanocomplexes (such as those shown in Figure 6A) could help select those that, like RAMOS, depend on the Ig β /CD19 complex for optimal fitness once BCR expression is lost.

As explained in detail above we now found the CD19/ Ig β nano-compartment also on a second tumor line and even on normal B cells (see new Fig. EV2 and Fig. EV3), a discovery that is in line with the literature on the special function of Ig β as discussed in our manuscript.

Minor points

- Figure 2A, 3A: provide MFI data for each mutant/complemented tumor population to better appreciate the expression levels of the corresponding proteins.

As suggested by reviewer 2 we now show the MFI data in the modified Fig.2 and Fig.3.

Referee #3:

Continuous signaling of CD79b and CD19 are required for the fitness of Burkitt lymphoma B cells

It is well established that BCR expression is essential for development as well as for the maintenance of mature B cells. Burkitt lymphoma requires continuous BCR signaling for their tumor growth. This is driven by ITAM and PI3K signaling.

The authors show, using CRISPER/Cas9 technologies to delete BCR as well as co-receptor genes in human BL cell line Ramos, that the competitive fitness of the BL cell line depends on the expression of Ig beta and CD19 and proper ITAM signaling.

The authors propose that Ig beta and CD19 are part of an alternative B cell signaling module that promotes the survival of BL cells and also normal B cells.

Further they show in this paper that in the absence of any BCR component Ig beta can be expressed on the surface close to CD19 and signals in an ITAM dependent manner.

With this data they claim that Ig beta and CD19 are part of an alternative B cell signaling module that use continuous ITAM/PI3K signaling to promote the survival of B cell lymphoma and normal B cells.

Comments:

The paper is well written and the experiments are of high quality and underline the presented theory.

We thank reviewer 3 for this positive statement.

Main critic:

The study relies completely on one cell line type and this cell line is derived from a cancer patient, namely these are not normal B cells. Although the results are very interesting, one remains wondering if this is the case also for "normal" B cells, as suggested by the authors.

There are many mouse lines with mutations in the BCR components, as well reviewed in the introduction by the authors. I would like the authors to repeat, at least their main findings with cells from mouse mutants that are similar to the mutations shown in this paper. One possible source of BCR deficient cells could be the system recently published by the groups of Rajewsky and Casola in Nature, where the mice express MYC and lack BCR expression. Although this system also makes use of malignant B cells, it would be interesting to use it to study whether in other systems Ig beta is expressed on the surface of the cells in the absence of the BCR.

*We agree with reviewer 3 that it would be important to show that the increased Ig β to CD19 proximity is not only seen in Ramos cells losing their BCR. We thus have repeated this analysis with another human Burkitt lymphoma line, namely DG75 and found the same phenotypes (see **new Fig. EV2**). According to the suggestion of reviewer 3 we have also studied “normal” murine splenic B cells that, after an inducible deletion of the VH gene, lose their IgH chain and BCR expression. In the **new Fig. EV3** we show that these cells maintain the Ig β /CD19 Fab-PLA signal although they lose the kappa staining and thus are BCR negative.*

Minor comments:

1. The authors should show their demonstrating that Ramos B cells lacking Ig alpha cannot respond by calcium flux to anti-Ig alpha, as they did for the Ig beta.

*As suggested, we are now showing in the **new Appendix Fig. S4** that upon exposure to anti-Ig α antibodies only WT but not BCR negative Ramos B cells flux calcium. This together with the anti-Ig β data shown in Fig.2 suggests that only Ig β but not Ig α can come on the cell surface in HL KO Ramos B cells.*

2. Fig. 3H, make sure the arrow is pointed correctly.

We have now corrected the position of the arrow in Fig. 3H.

3. Please quantify the levels of Ig beta in 3A and 3E, its hard to see differences when data is presented in these histograms.

*We now show the MFI data in the modified **Fig.2 and Fig.3**.*

4. It might be a problem of mac/PC, but on my computer some symbols appear as unknown signs. For example in the text of page 7. Please check.

2nd Editorial Decision

21st February 2018

Thank you for submitting your revised manuscript to The EMBO Journal. Your study has now been seen by referees #2 and 3 and their comments are provided below.

As you can see, both referees appreciate that the analysis has been strengthened. Referee #2 has some remaining issues that should be resolved in a final revision. I anticipate that you should be able to address them in a good way and most of them concern the need for a better description of how the experiments were done and data interpretation. No new issues have been brought up - they are all related to the initial review and the carried out revisions. Let me know if we need to discuss anything further

REFeree REPORTS

Referee #2:

In their revised manuscript, He and colleagues provide further evidence in support of an Ig β -CD19 nano-complex sustaining the in vitro competitive tumor growth of the malignant Burkitt lymphoma cell line RAMOS upon genetic extinction of surface BCR expression.

The manuscript has overall improved its quality. However, important details concerning the experimental setting and the interpretation of the data remain elusive and, hence, need further clarification.

Specifically:

- Clone representation in described experiments

In the revised manuscript, information is still missing on the number of independent clones for each mutant BCR genotype used to perform the various experiments indicated in Figures 1-to-6. Moreover, when authors refer to experiments performed at least three independent times, one wonders whether in each experiment different clones were used for each BCR genotype or always the same clones were employed.

- Effects on tumor cell fitness of inactivation of one or more BCR components

In response to the reviewer, authors include in the revised manuscript additional data related to the competitive growth properties of RAMOS cells mutant for one or more BCR components at a putative early time point after genetic inactivation (described in Figure S3). Authors indicate that this analysis was performed with clones (?). Were tumor cells first cloned and after that placed in competition?? If this was the case, it is likely that clones were kept in culture for prolonged time (at least 2-3 weeks) before any competition was started. What would happen if the experiment is performed on bulk sorted BCR mutant cells placed in competition soon after CRISPR/cas9 induced gene mutagenesis?? Including these data would be very helpful. In any case, the experiments shown in Figure S3 suggest that RAMOS mutant lacking Ig heavy chain (H KO) or IgL chain (L KO) are getting counter-selected overtime, although with slower kinetics in comparison to Ig β mutants. Applying some statistical analysis would help to assess whether the differences seen when WT cells compete with either WT or H KO or L KO B cells are significant.

- Growth properties of RAMOS cells upon inactivation of one or more BCR constituents

In Figures 1 and S2, authors provide convincing evidence that doubling time is by-and-large comparable between wt and BCR mutant RAMOS cells grown in isolation. This information does not allow to compare overall the growth properties of wt and BCR mutant lymphoma cells in vitro. Indeed, the lack of one or more BCR components may affect the survival (rather than the doubling time) of the lymphoma cells, possibly limiting their in vitro growth in isolation and/or under competitive settings. The manuscript would greatly benefit from showing cumulative growth curves of wildtype and BCR mutant tumor cultures kept in isolation and under competition for RAMOS and DG75 cells

- Extending/confirmation of the findings to other BL cells and WT B cells

The revised manuscript incorporates new data supporting the knowledge that Ig β /CD19 nanocomplexes may exist also in another BL cell line (DG75) and, further, also in resting primary mouse B cells. These interesting data will benefit from clarifying few critical points:

- DG75 cells:

- o information on the number of clones established for each BCR mutant genotype used for the analysis is missing
- o Surprisingly, no information was provided on the impact of inactivation of the various BCR components on the fitness of the tumor cells. Without this information, the possible functional relevance of Ig β /CD19 nano-complexes found in these cells remains intangible.

- Primary resting B cells: authors provide evidence for the existence of CD19/Ig β PLA-positive signals in these cells both before and after acute BCR ablation. Given that IgH ablation in the mouse model exploited by He and colleagues leads to the rapid disappearance of resting B cells, it remains unclear what function Ig β /CD19 complexes may possibly exert in these cells. Authors are encouraged to address this key point in the discussion.

In summary, whereas the manuscript provides compelling evidence for a role of the Ig β /CD19 complex in sustaining the competitive growth of RAMOS BL cells lacking BCR expression, several question marks remain open concerning the relevance of the findings in other BL cells (see comments listed above for DG75 cells) and in wild-type resting B cells.

Addressing the points listed above may help to better interpret the presented data and to justify the

conclusions. Also, complementing the revised manuscript with missing information on important technical details related to multiple experiments (as outlined above) is needed to fully appreciate the relevance of the data.

Referee #3:

The authors have answered to all of my concerns; the paper can be accepted as is now.

2nd Revision - authors' response

28th February 2018

Referee #2:

In their revised manuscript, He and colleagues provide further evidence in support of an Ig β -CD19 nano-complex sustaining the in vitro competitive tumor growth of the malignant Burkitt lymphoma cell line RAMOS upon genetic extinction of surface BCR expression.

The manuscript has overall improved its quality. However, important details concerning the experimental setting and the interpretation of the data remain elusive and, hence, need further clarification.

Specifically:

- Clone representation in described experiments

In the revised manuscript, information is still missing on the number of independent clones for each mutant BCR genotype used to perform the various experiments indicated in Figures 1- to 6. Moreover, when authors refer to experiments performed at least three independent times, one wonders whether in each experiment different clones were used for each BCR genotype or always the same clones were employed.

We now specify at each figure legend the clones of RAMOS KO cells used for the figure. We want to emphasize, that although in most of our figures we used only one clone of each genotype, we derived Ig β deficient clones by different routes (see Fig 1A and Fig 2A). Furthermore, we used batch sorted Ramos cells whenever it was applicable. For data presented in the Appendix figure S3 added during the revision, we generated completely new clones, and we observed same results as we have shown in Figure 1 and 2. In addition, we can reconstitute the function of the KO gene by re-expressing the corresponding gene. Taken together, we do not think our findings reported in this manuscript are clone specific.

- Effects on tumor cell fitness of inactivation of one or more BCR components

In response to the reviewer, authors include in the revised manuscript additional data related to the competitive growth properties of RAMOS cells mutant for one or more BCR components at a putative early time point after genetic inactivation (described in Figure S3). Authors indicate that this analysis was performed with clones (?). Were tumor cells first cloned and after that placed in competition? If this was the case, it is likely that clones were kept in culture for prolonged time (at least 2-3 weeks) before any competition was started. What would happen if the experiment is performed on bulk sorted BCR mutant cells placed in competition soon after CRISPR/cas9 induced gene mutagenesis?? Including these data would be very helpful. In any case, the experiments shown in Figure S3 suggest that RAMOS mutant lacking Ig heavy chain (H KO) or IgL chain (L KO) are getting counter-selected overtime, although with slower kinetics in comparison to Ig β mutants. Applying some statistical analysis would help to assess whether the differences seen when WT cells compete with either WT or H KO or L KO B cells are significant.

In this manuscript, we described that the fitness of RAMOS cells depends on the BCR independent Ig β signaling based on the finding that the HL β KO cells but not the HL KO cells or HL α KO cells were lost during the competition with WT RAMOS cells. Technically, it is hard to batch sort the HL KO cells from the RAMOS cells, since both the H and L single KO will lead to the loss of surface BCR already. It is also difficult to sort the HL α KO cells from the parental HL KO cells, since the HL KO cells has already lost the expressing of surface Ig α . The HL β KO cells indeed express less surface Ig β comparing with the HL KO cells, However, the rather small difference (see Figure 2B) makes it difficult to batch sort the cells. In addition, we do not always get high knockout efficiency when we use the CRISPR/cas9 method. For some gene, we had very low efficiency even after single cell sorting. It will be expected that we will have very limited number of cells after batch sorting and that means weeks of recovery before the competition assay.

Indeed, it seems that the data presented in the new Fig.S3 would suggest that the H KO and L KO RAMOS cells are also got counter selected. However, when we perform a multiple t-test comparing WT cells competing with WT cells to WT cells competing KO cells (WT:KO) at different days (each day is analyzed individually for t-test without assuming the same SD) and further analyze the multiple t-test results using the False Discovery Rate (FDR) approach with Two-stage linear step-up procedure (Benjamini, Y., Krieger, A. M. & Yekutieli, D. (2006) Adaptive linear step-up procedures that control the false discovery rate. *Biometrika* 93, 491–507.) by setting the desired FDR (Q) to 1 (Recommended value by the method), we found that for the time period of day 2 to day 8, significant differences (Q<0.01) were only seen when we compare the competition results between WT:null, or WT: β KO but not the WT: H KO or WT: L KO or WT: α KO supporting our notion that the competitive fitness of Ramos B cells is dependent on Ig β expression. See table for referee below for the detail. We have now also included this statistical analysis results in the modified Fig.S3.

Table for referee

WT:H KO	Discovery?	Q Value	P value	Mean 1	Mean 2	SE of difference	t ratio	df
Day 2	No	0,166149992	0,164504942	49,14	47,96	0,7477	1,583	6
Day 3	No	0,012146304	0,001718006	48,02	45,54	0,4612	5,366	6
Day 4	No	0,04534746	0,019422582	46,14	42,77	1,066	3,166	6
Day 5	No	0,04534746	0,031128914	44,71	39,96	1,695	2,801	6
Day 6	No	0,04534746	0,020376026	43,7	37,25	2,061	3,128	6
Day 7	No	0,066148307	0,056137177	40,91	34,08	2,895	2,362	6
Day 8	No	0,04534746	0,032070339	39,15	30,69	3,047	2,778	6
WT: L KO								
Day 2	No	0,523171353	0,517991438	49,14	48,5	0,9184	0,6951	5
Day 3	No	0,03277423	0,004635676	48,02	46,1	0,3948	4,859	5
Day 4	No	0,086138453	0,07310194	46,14	43,47	1,178	2,263	5
Day 5	No	0,071634471	0,048838087	44,71	39,69	1,939	2,59	5
Day 6	No	0,070401613	0,029873386	43,7	36,5	2,393	3,007	5
Day 7	No	0,071634471	0,050660871	40,91	31,96	3,5	2,56	5
Day 8	No	0,070401613	0,025941737	39,15	27,79	3,631	3,131	5
WT: α KO								
Day 2	No	0,814944655	0,806875897	49,14	48,89	0,973	0,2577	5
Day 3	No	0,099706049	0,014102694	48,02	45,86	0,585	3,693	5
Day 4	No	0,298438124	0,245160234	46,14	44,33	1,374	1,316	5
Day 5	No	0,220611542	0,085814904	44,71	40,34	2,049	2,135	5
Day 6	No	0,220611542	0,093611687	43,7	38,36	2,585	2,067	5
Day 7	No	0,298438124	0,249791717	40,91	36,3	3,546	1,302	5
Day 8	No	0,298438124	0,253271392	39,15	34,27	3,788	1,291	5
WT: Null								
Day 2	Yes	0,006473692	0,006409596	49,14	45,02	0,9167	4,498	5
Day 3	Yes	0,000301123	4,25917E-05	48,02	40,72	0,5476	13,32	5
Day 4	Yes	0,000867832	0,000490994	46,14	35,09	1,38	8,007	5
Day 5	Yes	0,000867832	0,000467624	44,71	28,84	1,961	8,09	5
Day 6	Yes	0,000867832	0,000336396	43,7	22,61	2,431	8,676	5
Day 7	Yes	0,001620912	0,001158139	40,91	17,74	3,484	6,652	5
Day 8	Yes	0,001620912	0,001375597	39,15	15,55	3,684	6,405	5
WT: β KO								
Day 2	Yes	0,001524918	0,010568741	49,14	46,05	0,7774	3,976	5
Day 3	Yes	4,14793E-05	4,10686E-05	48,02	40,72	0,5438	13,42	5
Day 4	Yes	0,000101285	0,00025776	46,14	33,99	1,324	9,176	5
Day 5	Yes	0,000101285	0,00040113	44,71	28	1,999	8,358	5
Day 6	Yes	0,000101285	0,000314164	43,7	22,68	2,388	8,802	5
Day 7	Yes	0,000180092	0,001069856	40,91	17,76	3,42	6,768	5
Day 8	Yes	0,000180092	0,000955201	39,15	13,75	3,661	6,938	5

• Growth properties of RAMOS cells upon inactivation of one or more BCR constituents
 In Figures 1 and S2, authors provide convincing evidence that doubling time is by-and-large comparable between wt and BCR mutant RAMOS cells grown in isolation. This information

does not allow to compare overall the growth properties of wt and BCR mutant lymphoma cells in vitro. Indeed, the lack of one or more BCR components may affect the survival (rather than the doubling time) of the lymphoma cells, possibly limiting their in vitro growth in isolation and/or under competitive settings. The manuscript would greatly benefit from showing cumulative growth curves of wildtype and BCR mutant tumor cultures kept in isolation and under competition for RAMOS and DG75 cells

We agree with referee #2 that the data presented in Fig1B and S2 only indicate the doubling time of the Ramos KO B cells lacking certain BCR components. However, the key finding of our manuscript is that the competitive fitness and not the proliferation of Ramos cells rely on Ig β expression and we think the competition data we provided in Fig.1E, 1G, 2C, 3B, 3C, 3F, 3G, as well as the newly added appendix figure S3 have clearly demonstrated that and we do not see the necessity to provide a cumulative growth curves for each of these experiments apart from those we show in Fig1B and S2.

- Extending/confirmation of the findings to other BL cells and WT B cells

The revised manuscript incorporates new data supporting the knowledge that Ig β /CD19 nanocomplexes may exist also in another BL cell line (DG75) and, further, also in resting primary mouse B cells. These interesting data will benefit from clarifying few critical points:

- DG75 cells:
 - o information on the number of clones established for each BCR mutant genotype used for the analysis is missing

We now include a statement for the clones of RAMOS KO cells used in all figure legends including that of the Expanded view figure 2 presenting the DG75 cell data.

- o Surprisingly, no information was provided on the impact of inactivation of the various BCR components on the fitness of the tumor cells. Without this information, the possible functional relevance of Ig β /CD19 nano-complexes found in these cells remains intangible.

It is not in the scope of this manuscript to repeat 3 years of work on the Ramos system with the DG75 cells. However, in the discussion of our manuscript we mention several in vivo studies whose result can now be better understood by our finding of a separated CD19/Ig β pro-survival signaling module and we surprised that referee #2 is not appreciating this.

- Primary resting B cells: authors provide evidence for the existence of CD19/Ig β PLA-positive signals in these cells both before and after acute BCR ablation. Given that IgH ablation in the mouse model exploited by He and colleagues leads to the rapid disappearance of resting B cells, it remains unclear what function Ig β /CD19 complexes may possibly exert in these cells. Authors are encouraged to address this key point in the discussion.

We think we have addressed the in vivo function of the CD19/Ig β pro-survival signaling module in our discussion.

In summary, whereas the manuscript provides compelling evidence for a role of the Ig β /CD19 complex in sustaining the competitive growth of RAMOS BL cells lacking BCR expression, several question marks remain open concerning the relevance of the findings in other BL cells (see comments listed above for DG75 cells) and in wild-type resting B cells.

Addressing the points listed above may help to better interpret the presented data and to justify the conclusions. Also, complementing the revised manuscript with missing information on important technical details related to multiple experiments (as outlined above) is needed to fully appreciate the relevance of the data.

Corresponding Author Name: Michael Reth

Journal Submitted to: the EMBO Journal

Manuscript Number: EMBOJ-2017-97980R